# LUX ARRHYTHMO mediates crosstalk between the circadian clock and defense in Arabidopsis

Chong Zhang[1,5,6], Min Gao[1,6], Nicholas C. Seitz[1], William Angel[1], Amelia Hallworth[1], Linda Wiratan [1],
Omar Darwish[2], Nadim Alkharouf[2], Teklu Dawit[1], Daniela Lin[1], Riki Egoshi[1], Xiping Wang[3],
C. Robertson McClung [4] & Hua Lu [1]

The circadian clock is known to regulate plant innate immunity but the underlying mechanism of this regulation remains largely unclear. We show here that mutations in the core clock component LUX ARRHYTHMO (LUX) disrupt circadian regulation of stomata under free running and *Pseudomonas syringae* challenge conditions as well as defense signaling mediated by SA and JA, leading to compromised disease resistance. RNA-seq analysis reveals that both clock- and defense-related genes are regulated by LUX. LUX binds to clock gene promoters that have not been shown before, expanding the clock gene networks that require LUX function. LUX also binds to the promoters of *EDS1* and *JAZ5*, likely acting through these genes to affect SA- and JA-signaling. We further show that JA signaling reciprocally affects clock activity. Thus, our data support crosstalk between the circadian clock and plant innate immunity and imply an important role of *LUX* in this process.

[1] Department of Biological Sciences, University of Maryland Baltimore County, 1000 Hilltop Circle, Baltimore, MD 21250, USA. [2] Department of Computer and Information Sciences, Towson University, Towson, MD 21252, USA. [3] State Key Laboratory of Crop Stress Biology in Arid Areas, College of Horticulture, Northwest A & F University, 712100 Yangling, Shaanxi, China. [4] Department of Biological Sciences, Dartmouth College, Hanover, NH 03755, USA. [5] Present address: Genetic Improvement of Fruits and Vegetables Laboratory, USDA-ARS, Beltsville, MD 20705, USA. [6] These authors contributed equally: Chong Zhang, Min Gao. Correspondence and requests for materials should be addressed to H.L. (email: hualu@umbc.edu)

In response to various pathogens and pests, plants have evolved sophisticated defense mechanisms to recognize and fight these invaders. One such mechanism uses the internal time measuring machinery, the circadian clock, to modulate defense responses in anticipation of pathogens and pests at the time of day when they are likely to be encountered as well as during an actual attack[1]. The circadian clock is known to have a profound influence on plant growth, development, and responses to environmental cues[2–4], although the mechanisms by which the circadian clock regulates plant defense are only beginning to be elucidated.

Although the molecular composition of the circadian clock differs greatly between plants and other organisms, the basic principle of clock function, which is the ability to self-sustain an approximately 24-h cycle, is conserved. Like in other organisms, the circadian clock in plants consists of core clock components, which form complicated interlocking transcription–translation feedback loops (TTFLs) that are subject to both transcriptional and posttranscriptional regulation[2–4]. Core clock genes are expressed at different times of a day and can affect the expression and/or activities of each other as well as of genes acting in output pathways. The concerted function of these clock genes calibrates the circadian clock and keeps timing in a precise, self-sustaining manner. For instance, the two homologous Myb transcription factors, CIRCADIAN CLOCK-ASSOCIATED 1 (CCA1) and LATE ELONGATED HYPOCOTYL (LHY), are expressed in the morning (morning-phased) and contribute to multiple clock TTFLs through a direct regulation of several other core clock genes and themselves. CCA1 and LHY also directly regulate the expression of many clock output genes[5]. One target of the CCA1 protein is the evening-phased core clock gene LUX ARRHYTHMO (LUX), also known as PHYTOCLOCK1, which encodes a GARP transcription factor essential for circadian rhythmicity[6,7]. Expression of LUX is also affected by several other clock proteins, including TIMING OF CAB EXPRESSION 1 (TOC1), REVEILLE 8 (RVE8), PSEUDO-RESPONSE REGULATOR 5 (PRR5), and PRR7[8–11]. In turn, LUX binds directly to the conserved LUX-binding site (LBS) in the promoters of several clock genes, including GIGANTEA (GI), NIGHT LIGHT-INDUCIBLE AND CLOCK-REGULATED GENE 1 (LNK1), PRR7, PRR9, and LUX itself, to regulate their expression[7,12]. Thus, like CCA1 and LHY, LUX is involved in multiple clock TTFLs. LUX, at least in part, functions through interactions with other proteins. LUX or its close homolog, BROTHER OF LUX ARRHYTHMO (BOA), forms the evening complex (EC) with two evening-phased proteins, EARLY FLOWERING 3 (ELF3) and ELF4[13,14]. The EC affects many aspects of plant development and physiology, including growth, flowering, and cold response, as clock outputs[15].

Recent studies have demonstrated a critical role of the circadian clock in plant defense against pathogens and pests. Disruption of certain clock genes leads to reduced resistance against bacteria, oomycete, and/or fungal pathogens[1]. Arrhythmicity caused by misexpressing LUX or CCA1 compromises insect resistance[16]. The temporal control of defense by the circadian clock manifests in the rhythmic changes of defense-related molecules, reflecting the role of the circadian clock in anticipating likely attacks from pathogens and pests. For instance, in the absence of pathogens and pests, expression of many defense-related genes and production of defense signaling molecules, such as salicylic acid (SA), jasmonic acid (JA), and reactive oxygen species (ROS), oscillate with varying peaks during the day[16–19]. However, in the presence of pathogens and pests, plants activate acute defense responses, including drastic increases in SA and other defense compounds and reprogramming of defense-related genes. Most of these acute responses lose the rhythmic signature observed under the unchallenged condition. For instance, while the levels of SA oscillate daily in unchallenged plants[16,17], timely accumulation of SA in high abundance in the local infected region dictates the outcome of plant response to some pathogens[20,21]. Genes affecting such acute SA accumulation are important for plant defense[22–24], although no clock genes have yet been reported to play such a role in SA regulation. Thus how the circadian clock gates acute defense responses in the presence of pathogens and pests remains largely unknown.

In order to identify circadian clock genes that contribute to SA regulation, we conducted a genetic analysis with a unique Arabidopsis mutant, acd6-1, which exhibits constitutively high levels of defense that are inversely correlated with the size of the plant[25,26]. This feature of acd6-1 has proven useful in gauging the effects of potential mutations on defense[21,22,27–31]. We report here that lux-1, a nonsense mutation in the early coding region of the LUX gene[6], suppresses acd6-1-conferred dwarfism and high SA defense phenotypes. We confirmed the SA regulatory role of LUX with Pseudomonas syringae infection and further discovered a role of LUX in regulating JA signaling. This function of LUX arises, at least in part, through a direct control of the key SA and JA signaling genes, ENHANCED DISEASE SUSCEPTIBILITY (EDS1)[27,32] and JASMONATE ZIM-DOMAIN 5 (JAZ5)[33,34], respectively. LUX also affects temporal stomatal opening and closure under free running and acute pathogen challenging conditions. Consistent with the multiple functions of LUX in defense regulation, lux-1 is compromised in resistance to a broad spectrum of pathogens and pests. RNA-seq analysis followed by chromatin immunoprecipitation (ChIP) experiments supports a central role of LUX in clock and defense regulation. In addition, we show that activation of JA signaling affects LUX expression and reciprocally regulates clock activity. Together, our data reveal an important role of LUX mediating the crosstalk between the circadian clock and plant innate immunity.

## Results

**LUX regulates SA-mediated defense.** In order to identify circadian clock genes that gate plant defense, especially SA-mediated defense, we introduced several clock mutations into acd6-1, an Arabidopsis mutant with constitutive defense whose size is roughly inversely proportional to SA levels. We found that, while mutations in CCA1 and LHY did not affect acd6-1 size[35], the lux-1 mutation significantly suppressed acd6-1 dwarfism (Fig. 1a, b). Compared with acd6-1, acd6-1lux-1 also displayed decreased cell death, SA accumulation, expression of the defense marker gene PR1, and resistance to the virulent P. syringae pv. maculicola ES4326 strain DG3 (PmaDG3) (Fig. 1c–f). The Col-0 and lux-1 plants appeared largely similar in their morphology except that lux-1 had slightly longer petioles. These results suggest a role of LUX in regulating SA-mediated defense.

To further confirm this role of LUX, we challenged Col-0 and lux-1 plants with PmaDG3 and collected the infected leaves in a time course for SA quantification (Fig. 1g). We found that there was a significant reduction in SA levels 16 and 20 h post infection (hpi) in lux-1, as compared to Col-0. Thus these data support a role of LUX in regulating acute SA accumulation in local tissue upon P. syringae infection.

**LUX regulates multiple layers of defense responses.** The SA regulatory role of LUX suggests that LUX is important for plant disease resistance. Consistent with this idea, we found that expression of LUX was induced by infection with P. syringae strains, the virulent PmaDG3 and the avirulent strain Pma avrRpm1 (Supplementary Fig. 1). To further establish a role of LUX in defense regulation, we grew plants in a chamber with 12 h

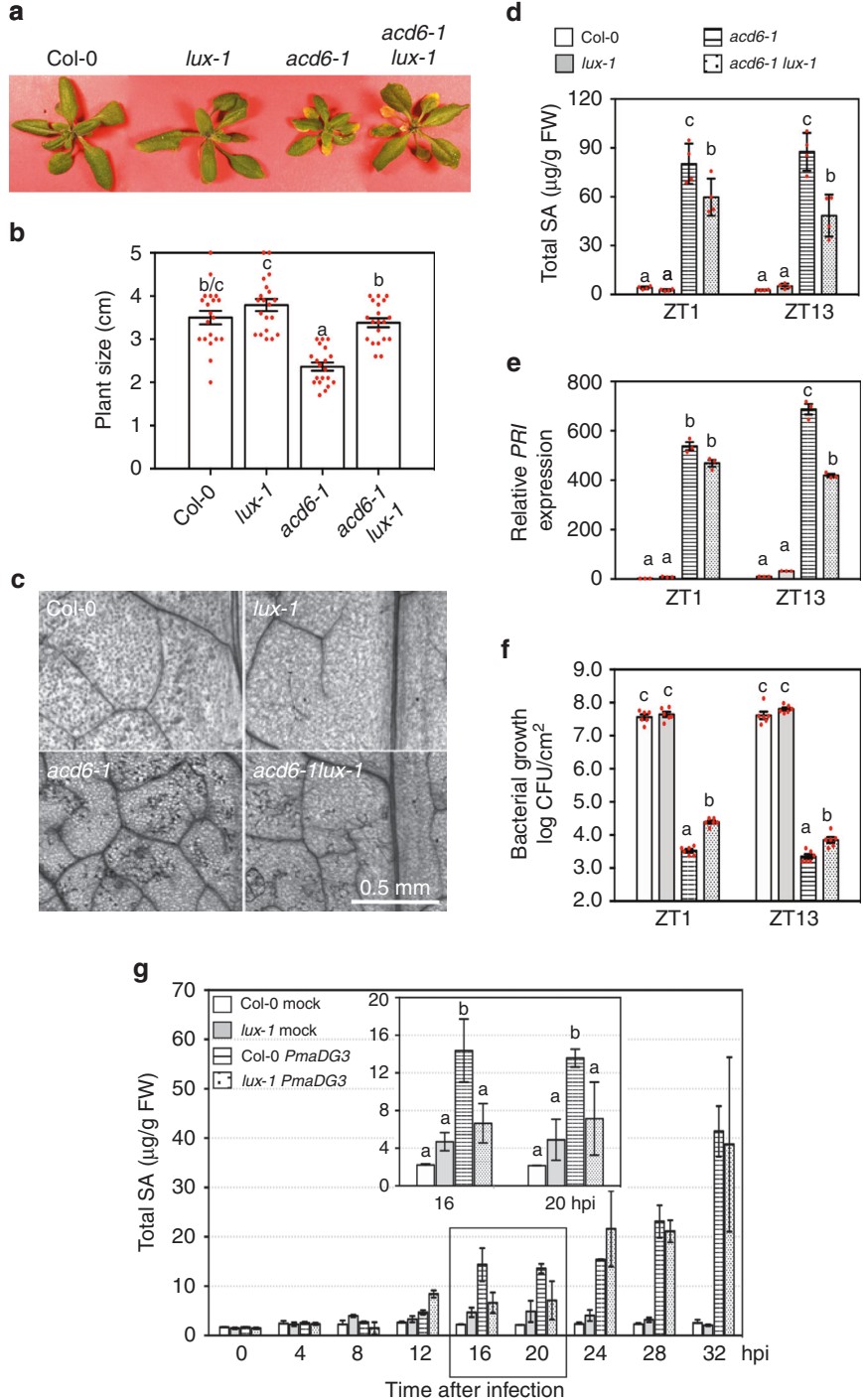

**Fig. 1** The *lux-1* mutation suppresses salicylic acid (SA)-mediated defense. **a** Phenotypes of 25-day-old plants. **b** Average size of 25-day-old plants. Plants were measured for the largest distance between tips of two rosette leaves (*n* = 20). **c** Cell death staining of the fifth to seventh leaves of plants. **d** SA quantification. Whole plants were collected at ZT1 or ZT13 for SA extraction followed by high-performance liquid chromatography measurement (*n* = 4 from two independent experiments). **e** Expression of *PR1*. Whole plants of each genotype were collected at ZT1 or ZT13 for RNA extraction followed by quantitative reverse transcriptase–PCR (qRT-PCR) analysis (*n* = 3). **f** Bacterial growth. The fourth to sixth leaves of each genotype were infiltrated with *PmaDG3* (OD = 0.0001) at ZT1 or ZT13 and assessed for bacterial counts at 3 dpi (*n* = 6). **g** SA quantification with plants infected by *PmaDG3*. The fourth to sixth leaves of each genotype were infiltrated with *PmaDG3* (OD = 0.01) or the mock solution at ZT1 and collected at the indicated time points for SA analysis (*n* = 2). Data represent mean ± SD in **d**, **e**, **g** and mean ± SEM for **b**, **f**. Statistical analysis was performed with one-way analysis of variance with post hoc Tukey honestly significant difference test. Different letters in **b**, **d**–**g** indicate significant difference among the samples at the same time point (*P* < 0.05). These experiments were repeated three times with similar results

light/12 h dark (LD) and 180 µmol m$^{-2}$ s$^{-1}$ light intensity for 25 days and spray-infected the plants with *Pma*DG3 at zeitgeber time 1 (ZT1; ZT1 is 1 h after lights on) or ZT13 (1 hour after lights off). Infected plants were kept in continuous light (LL; a free running condition) and assessed for disease symptoms and bacterial growth. The infected *lux-1* leaves displayed more chlorosis than those of Col-0 but did not show increased bacterial growth 4 days post infection (dpi) (Supplementary Fig. 2a). The increased chlorosis in *lux-1* is consistent with the role of *LUX* as a repressor of leaf senescence[36]. High light intensity exacerbates the senescence phenotype in *lux-1*, complicating plant defense responses. To better assess the role of *LUX* in defense regulation, we lowered the light intensity from 180 to 10 µmol m$^{-2}$ s$^{-1}$ photon flux density in LL during infection (Fig. 2a). Under these conditions, Col-0 showed time-of-day-dependent defense, depending on the mode of *P. syringae* infections[35]. To sprayed *Pma*DG3, Col-0 was more susceptible at LL25 (subjective morning) than at LL37 (subjective evening) (Fig. 2b and ref. [35]). Interestingly, while the two *lux* mutants showed greater bacterial growth and more chlorosis than Col-0 with both LL25 and LL37 infections, *lux* demonstrated higher sensitivity to *Pma*DG3 in the morning than at night (Fig. 2b). To infiltrated *Pma*DG3, Col-0 was more susceptible at night than in the morning (Fig. 2c and refs. [35,37,38]). Such time-dependent susceptibility was abolished in *lux*, which demonstrated similar *Pma*DG3 growth when infected at both LL25 and LL37 but more bacterial growth than Col-0 at LL25 (Fig. 2c). Expressing the wild-type *LUX* gene translationally fused to the *GFP* reporter in *lux-4* (*LUX-GFP*)[6,39] rescued *lux*-conferred *Pma*DG3 susceptibility with infection at both LL25 and LL37 (Fig. 2b, c).

Because sprayed *Pma*DG3 gains access to the interior of plant tissue via stomata, we examined whether LUX affects stomata-dependent defense. In LL, Col-0 and *LUX-GFP* plants showed circadian-regulated stomatal aperture, with stomata more open in the morning than at night (Fig. 2d and Supplementary Fig. 2b). Consistent with them being arrhythmic, the *lux* mutants lost this temporal modulation of stomatal activity and kept stomata open at both LL25 and LL37. When being challenged with *Pma*DG3 in LL, Col-0 and *LUX-GFP* plants demonstrated a temporal stomatal response (Fig. 2d and Supplementary Fig. 2b). At LL25, 1 h *Pma*DG3 incubation transiently induced stomatal closure in Col-0 and *LUX-GFP* plants. At LL37, the stomatal aperture of Col-0 and *LUX-GFP* plants did not change within 3 h of exposure to *Pma*DG3. In contrast to Col-0 and *LUX-GFP*, the *lux* mutations disrupted this temporal response of stomata to *Pma*DG3 challenge. At LL25, the stomatal aperture of *lux* incubated for 1 h with *Pma*DG3 was smaller than that of mock-treated *lux* but was still significantly larger than that of *Pma*DG3-infected Col-0. At LL37, stomata of *lux* responded strongly to *Pma*DG3, showing significantly smaller aperture at 1 hpi than those of *lux* at 1 hpi at LL25. Open stomata observed in the *lux* mutants at LL37 could allow *Pma*DG3 to enter and thereby infect plant tissue, making the *lux* plants more susceptible than Col-0. The higher sensitivity of *lux* stomata to acute *Pma*DG3 infection at night than in the morning explains why the *lux* mutants showed more resistance at night than in the morning. Together, our data suggest that LUX regulates temporal defense in both stomata-dependent and stomata-independent pathways.

We further tested whether *LUX* is involved in activating defense signaling. Recognition of a pathogen avirulence effector by its cognate resistance protein in the local infected tissue often results in enhanced and durable resistance in distal uninfected regions, termed systemic acquired resistance (SAR). To test whether *LUX* affects SAR, we infiltrated at ZT1 the fourth to sixth leaves of Col-0, *lux-1*, and *lux-4* plants with *Pma avrRpt2* to activate SAR or with a mock solution as a control. At ZT1 2 dpi,

we challenged the plants with a secondary infection of *Pma*DG3. As expected, SAR-activated Col-0 plants showed more resistance to *Pma*DG3, compared with non-SAR-activated Col-0 (Fig. 2e). In contrast, no difference was observed in the *lux* mutants with the initial mock or *Pma avrRpt2* treatment. These results suggest that LUX is necessary for SAR activation.

To test whether *LUX* affects basal defense, we used flg22, a 22 aa peptide from the conserved region of the flagellin protein of *P. syringae*, to elicit basal immunity, also known as pathogen-associated molecular pattern-triggered immunity (PTI)[40]. We assayed two physiological changes associated with flg22-PTI, seedling growth inhibition and callose deposition[40,41]. We found that the *lux-1* mutant was less sensitive than Col-0 in flg22-induced seedling growth inhibition and callose deposition (Fig. 2f, g). We further challenged plants with another basal defense inducer, the *Pma hrcC$^{-}$* strain that lacks the type III secretion system to deliver effectors to plant cells[20]. We found that *Pma hrcC$^{-}$* induced significantly fewer callose deposits in *lux-1* than in Col-0 (Fig. 2g). Together, these data suggest that *LUX* contributes to basal defense in Arabidopsis.

**A role of *LUX* in regulating SA- and JA-mediated defense.** To elucidate how *LUX* is mechanistically linked to plant defense, we performed RNA-seq analysis. We generated transcriptome profiles of Col-0, *lux-1*, *acd6-1*, and *acd6-1lux-1* at both ZT1 and ZT13 with the goal of identifying potential LUX target genes. After removing low-quality reads, an average of 79.8% of filtered reads were mapped to the genome sequence of Arabidopsis (Supplementary Fig. 3a). Correlation dendrogram analysis indicated that all biological replicates of each sample clustered together. Expression profiles of the samples separated into two major groups based on *acd6-1* or Col-0 background (Supplementary Fig. 3b). In each group, the expression profile further clustered by time followed by the *LUX* genotype. These observations suggest that defense activation by *acd6-1* influences the global transcriptomic profile more profoundly than either time of day or the circadian clock gene *LUX*.

To identify *LUX*-affected genes, we compared four groups: (a) Col-0 vs. *lux-1* at ZT1; (b) Col-0 vs. *lux-1* at ZT13; (c) *acd6-1* vs. *acd6-1lux-1* at ZT1; and (d) *acd6-1* vs. *acd6-1lux-1* at ZT13. Table 1 shows that the number of genes affected by *lux-1* was generally higher in the day (groups a and c) than at night (groups b and d). There were more downregulated genes in the morning (groups a and c) and more upregulated genes in the evening (groups b and d). Under non-defense activation conditions (the Col-0 background; groups a and b), there were 790 genes differentially affected by *lux-1* at ZT1 and ZT13, less than the number of genes (1180) affected under defense activation conditions (the *acd6-1* background; groups c and d) (Supplementary Fig. 4a, b). A total of 1618 genes was found to be differentially affected by *lux-1* in at least one of the comparison groups (Table 1 and Supplementary Fig. 4c). Gene Ontology (GO) analysis of the *LUX*-affected genes revealed an enrichment of genes responding to abiotic and biotic stimuli in each comparison group and in all four groups combined, compared with the genome-wide gene expression profile (Table 1).

Cluster analysis of the 1618 *LUX*-affected genes revealed three major groups (Fig. 3a). Expression of many genes in group II was relatively low in all four genotypes, compared with expression in groups I and III. Some genes in group II showed greater expression in *lux-1*, consistent with the known role of LUX as a transcriptional repressor. Most genes in group I were highly induced in *acd6-1*. Expression of most group I and III genes was suppressed by *lux-1*, especially at ZT1 and/or in the *acd6-1* background, suggesting that LUX can also positively affect gene

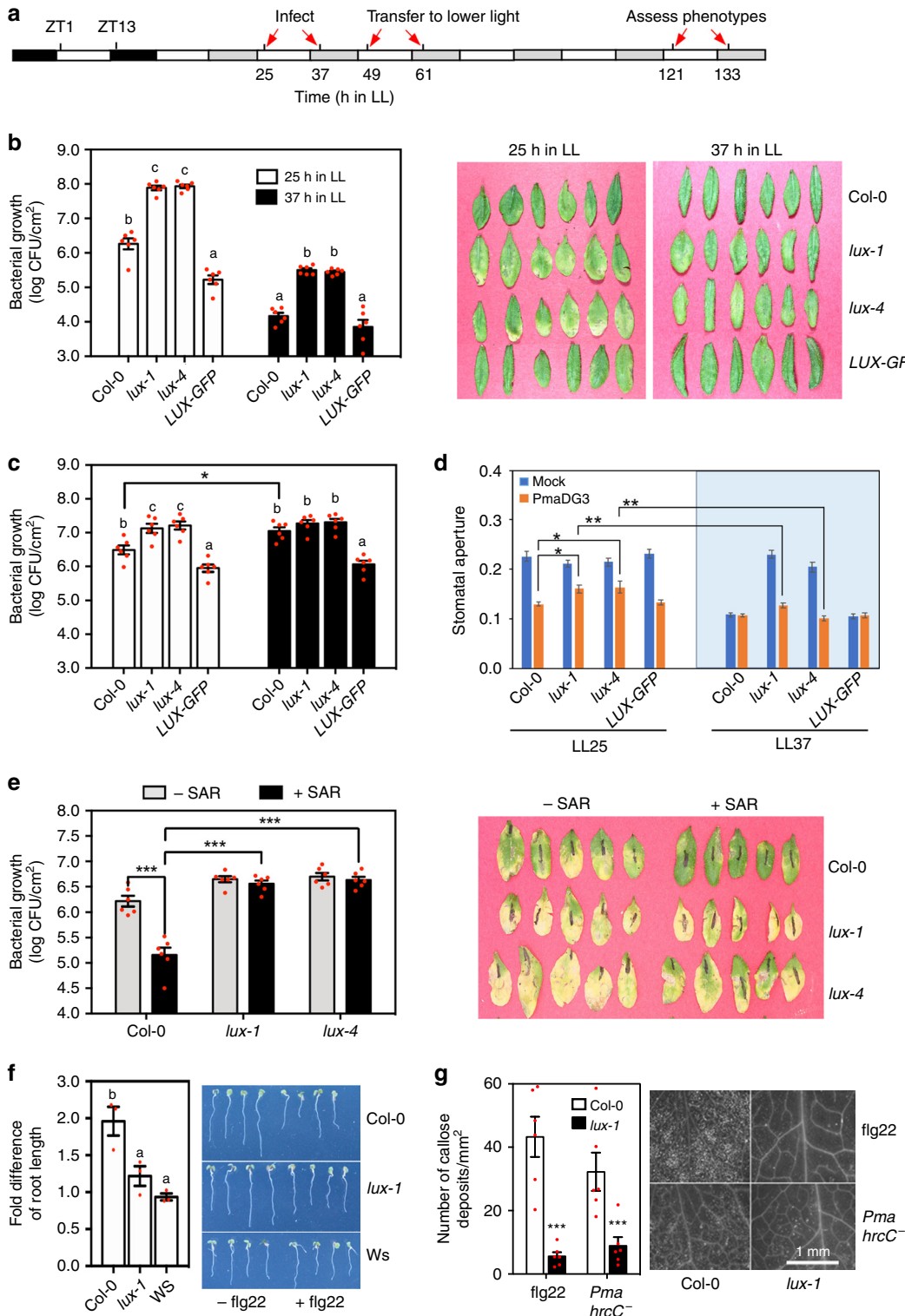

expression via direct or indirect means. GO analysis revealed that groups I and III were more enriched than group II in genes responding to abiotic and biotic stimuli (Fig. 3a). Among the stress-related genes, expression of several PTI and SA genes was downregulated in *lux-1*. Examples of these genes include PTI signaling genes, such as *MPK3*, *WRKY22*, and *WRKY33*, and SA-associated genes, such as the major SA biosynthesis gene *ICS1*, the SA regulatory genes *EDS1*, *PAD4*, and *PBS3/WIN3/GDG12*, the SA receptors *NPR1* and *NPR3*, and the SA marker gene *PR1*. Quantitative reverse transcriptase–PCR (qRT-PCR) analysis

confirmed expression of some of these genes (Supplementary Fig. 5). Together, these data support a positive role of *LUX* in regulating both PTI- and SA-mediated defense. We also noticed altered expression of a number of genes related to JA signaling in *lux-1*, including several JA regulatory genes (e.g., *JAZ1*, *JAZ5*, *JAZ7*, and *JAZ8*) and JA marker genes (e.g., *PDF1;2*, *VSP1*, and *VSP2*) (Fig. 3a and Supplementary Fig. 5). These observations support a role of *LUX* in regulating JA signaling.

The LUX protein has been shown to bind to the LBS motif in some clock gene promoters[7,12]. In order to uncover additional

**Fig. 2** *LUX* regulates stomata aperture and defense signaling. **a** Light and time scheme used for some treatments. 25-day-old plants grown in LD were transferred to LL for 1 day followed by pathogen infection 25 or 37 h after onset of LL. The light intensity was lowered from 180 to 10 $\mu$mol m$^{-2}$ s$^{-1}$ photon flux density at 49 h LL or 61 h LL for plants infected at 25 h LL or 37 h LL, respectively. **b** Bacterial growth (left) and photographs (right) of plants sprayed with *Pma*DG3 (OD$_{600}$ = 0.1) at LL25 or LL37 ($n$ = 6). **c** Bacterial growth of plants infiltrated with *Pma*DG3 (OD$_{600}$ = 0.0001) at LL25 (white bars) or LL37 (black bars) ($n$ = 6). **d** Stomatal aperture measured at 1 hpi of *Pma*DG3 (OD$_{600}$ = 0.1) at LL25 or LL37 ($n$ = 80). **e** Systemic acquired resistance (SAR) assay ($n$ = 6). SAR was induced by the primary infection of *Pma avrRpt2* (OD = 0.05). Bacterial growth (left) and disease symptom (right) was recorded with the secondary infection of *Pma*DG3 (OD = 0.0001) at 3 dpi in LD. **f** Seedling growth inhibition assay with 1 $\mu$M flg22. The fold difference was calculated as the ratio of water-treated root length/flg22-treated root length of each genotype. Data (left) represent the average of three independent experiments ($n$ = 4 per genotype/treatment in each experiment). Pictures of seedlings (right) were from one representative experiment. **g** Quantification (left) and images (right) of callose deposition with flg22 (1 $\mu$M) or *Pma hrcC*$^{-}$ (OD$_{600}$ = 0.1) treatment ($n$ = 6). Numbers of callose deposits were quantified using ImageJ (version 1.45). H$_2$O treatment did not induce callose deposition (not shown). Data represent mean ± SEM. Except **f**, other experiments were repeated two times and similar results were obtained. Statistical analysis was performed with one-way analysis of variance with post hoc Tukey honestly significant difference test. Significant difference between Col-0 and other plants at the same time point and/or with the same treatment was indicated by different letters or by asterisks (one asterisk for $P < 0.05$, two for $P < 0.01$, and three for $P < 0.001$)

---

**Table 1 Biotic and abiotic stress-related genes are preferentially regulated by LUX**

|  | Differentially expressed >2× | Upregulated | Downregulated | Response to abiotic and biotic stimuli |
|---|---|---|---|---|
| Col-0 vs. *lux-1* ZT1 (a) | 661 | 285 | 376 | 13.0% |
| Col-0 vs. *lux-1* ZT13 (b) | 170 | 127 | 43 | 17.4% |
| *acd6-1* vs. *acd6-1lux-1* ZT1 (c) | 1068 | 311 | 757 | 12.7% |
| *acd6-1* vs. *acd6-1lux-1* ZT13 (d) | 143 | 100 | 43 | 15.6% |
| a + b + c + d | 1618 | — | — | 11.1% |
| Whole genome genes | — | — | — | 5.4% |

The number of genes whose expression was affected in *lux-1* relative to Col-0 more than two-fold is listed for each comparison group. Gene Ontology annotations were used to assign genes into functional groups[72]

---

direct LUX targets, we identified the LBS motifs in both sense and antisense directions within 1500 bp upstream of the transcription start site of selected LUX-affected defense genes (Supplementary Table 1). We conducted chromatin immunoprecipitation (ChIP) with these gene promoters, using the *LUX-GFP* plants and Col-0 plants expressing the luciferase reporter driven by the *CAB2* gene promoter (*CAB2:LUC*) as a control[39]. Among the defense gene promoters tested, we detected LUX binding to the LBS motifs in the promoters of the SA regulator *EDS1* and the JA regulator *JAZ5* (Fig. 3b, c). The lack of LUX binding to the regions distal to the LBS motif in these gene promoters validated the specificity of our ChIP experiments. Thus *LUX* may act, at least in part, through *EDS1* and *JAZ5* in regulating SA and JA signaling, respectively.

To further test the role of *LUX* in JA signaling, we treated *lux-1* and Col-0 seedlings with methyl jasmonate (MJ) and found that *lux-1* was less sensitive to MJ than Col-0 in the root growth assay (Fig. 4a). Under *Pma*DG3 challenge, the *lux* mutant accumulated much higher levels of *JAZ5*, *MYC2*, *VSP1*, and *VSP2* transcripts, supporting a repressor role of *LUX* in regulating JA-related gene expression (Fig. 4b). Interestingly, the expression of *EDS1* was lower in *lux-1* upon *Pma*DG3 infection, suggesting the possibility that LUX acts as a transcriptional activator for *EDS1*.

*EDS1* was shown previously to function in both SA-dependent and SA-independent pathways[27,32,42]. We found that, like *lux-1*, *eds1-2* was less sensitive than Col-0 to MJ treatment in the root growth assay (Fig. 4a). Expression of *MYC2*, *JAZ5*, *VSP1*, and *VSP2* was more induced in *eds1-2* than in Col-0 upon *Pma*DG3 infection (Fig. 4c). These results indicate a role of *EDS1* in negatively regulating JA gene expression.

JA signaling is known to be important for plant defense against necrotrophic pathogens, such as *Botrytis cinerea*. To test whether *LUX* regulates broad-spectrum disease resistance, we spray-infected whole Col-0 and *LUX* misexpressing plants with *Botrytis* at LL25 or LL37. We found that the *lux* mutants showed more

severe necrotic lesions on the leaves than Col-0 and *LUX-GFP* plants with both infections (Fig. 4d). Similarly, the loss-of-function mutant of the *LUX*-interactor gene *ELF3*, *elf3-7*, also showed enhanced susceptibility to *Botrytis* at both times, consistent with the previous result of the *Botrytis* assay with detached leaves[43]. Thus both LUX and its interacting protein ELF3 are important for plant resistance to the fungal pathogen *Botrytis*. Interestingly, unlike the *lux* mutants, *eds1-2* was not compromised in its response to *Botrytis* infection (Fig. 4d). Together, our data suggest that *LUX* affects both SA- and/or JA-mediated signaling, ultimately influencing defense outcome in Arabidopsis.

**An expanded role of *LUX* in regulating clock TTFLs and output**. To find *LUX*-affected genes that also oscillate in expression throughout the day, the web-based tool Phaser[44,45] was used to analyze gene expression in LD and LL, using publicly available microarray data[46,47]. Of the 1618 *LUX*-affected transcripts, 26.7% cycled under LD and 26.3% cycled under LL. When we analyzed the entire Arabidopsis transcriptome, we found that 18.9% cycled in LD and 17.8% in LL. This observation of enrichment of cycling transcripts in the set of LUX-affected transcripts suggests that *LUX* preferentially regulates the expression of cycling genes, consistent with *LUX* being a core clock regulator.

Consistent with LUX being a transcription repressor, expression of many clock-regulated genes was higher in *lux-1*. These genes include core clock genes (e.g., *BOA*, *ELF4*, *GI*, *LNK1*, *LNK2*, *PRR5*, *PRR7*, *PRR9*, and *LUX* itself) and clock output genes that regulate plant growth (*PIF4*)[39], flowering time control (*CDF1*, *CO*, *FLC*, and *FT*)[48,49], and cold response (*CBF1*, *CBF2*, and *CBF3*)[50]. These genes are expressed with peaks at multiple and distinct times of day[9], suggesting that the evening clock gene *LUX* has a profound influence on global gene expression at different times of day. It is important to note that, although known as an arrhythmic mutant in LL, *lux-1* shows robust driven rhythms in

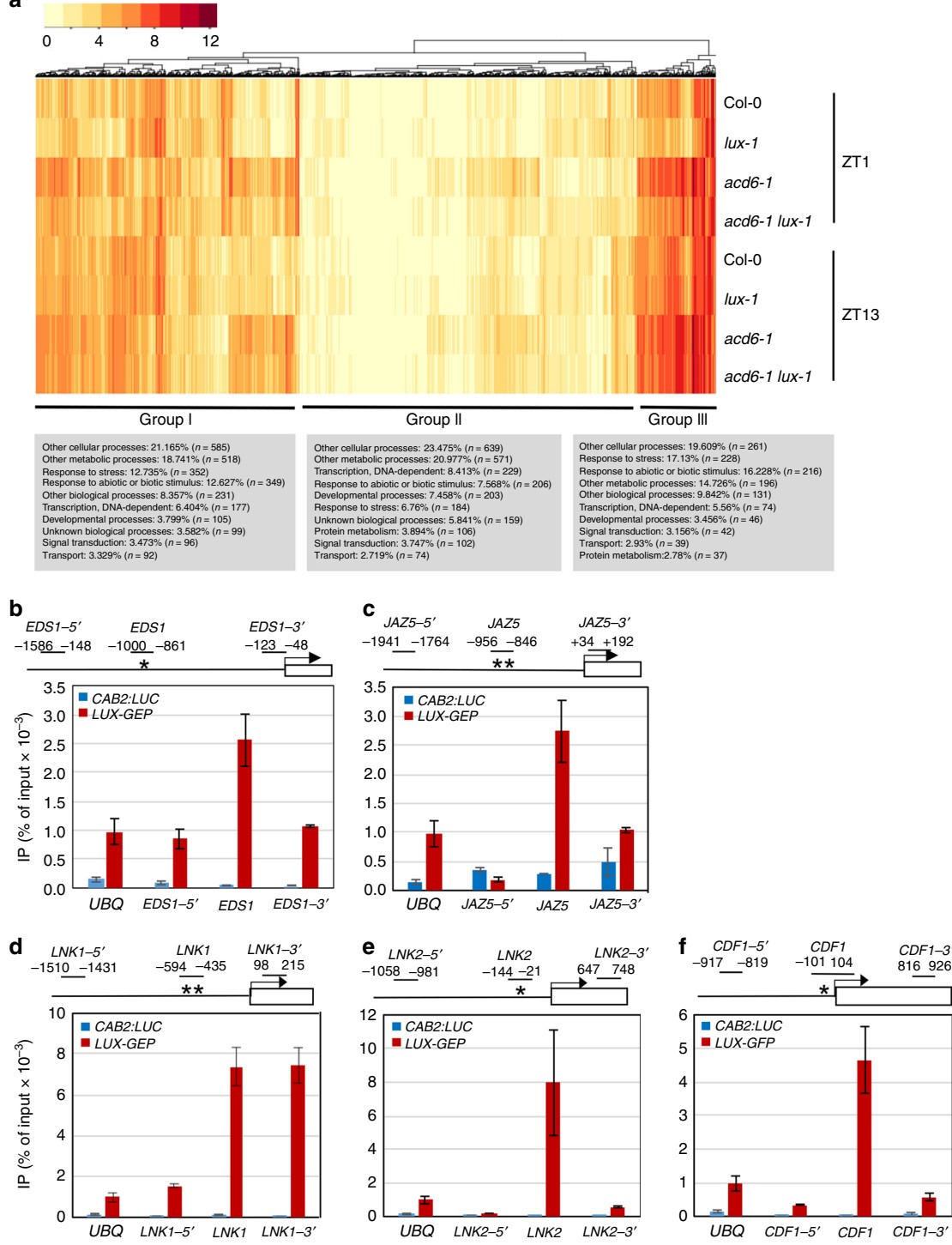

**Fig. 3** Elucidation of *LUX* target genes via RNA-seq and chromatin immunoprecipitation (ChIP) experiments. **a** Heatmap analysis of 1618 *LUX*-affected genes. Relative RPKM values of *LUX*-affected genes were used in the cluster analysis, using the heatmap.2() function in the R package gplots. **b–f** ChIP experiments detect in vivo association of LUX with the promoters of clock and defense genes. ChIP assays were performed with *LUX-GFP* plants or Col-0 expressing *CAB2:LUC* as the negative control[39]. Anti-GFP antibodies were used for IP, using mock treatment as a control. Each amplicon from a specific primer pair at the indicated position (relative to the transcription start site) in each gene promoter was quantified after normalization with the internal control (*UBQ*). Asterisks indicate the predicted LUX-binding site within the 1500-bp promoter region upstream of the transcription start site of each gene. Data represent mean (*n* = 2) ± SD. This experiment was repeated four times for *EDS1* and *JAZ5* and two times for *LNK1*, *LNK2*, and *CDF1* and similar results were obtained

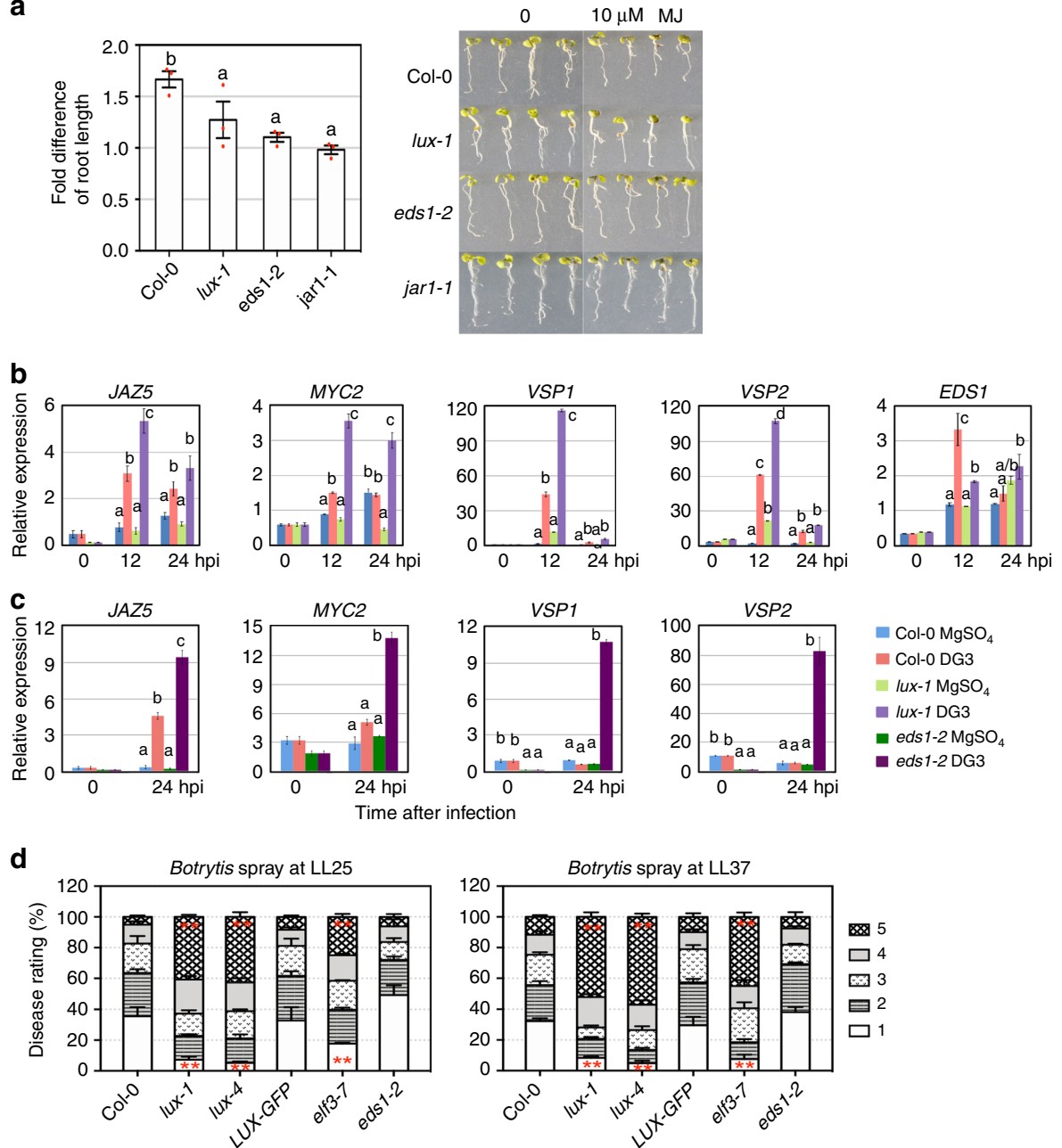

**Fig. 4** Mutations in *LUX* and *EDS1* affect jasmonic acid response and signaling. **a** Root length inhibition with methyl jasmonate (MJ) treatment. The fold difference of seedling root length (left) was calculated as the ratio of seedling root length with water treatment/seedling root length with 10 μM MJ treatment. The MJ-insensitive mutant *jar1-1*[68] was used as a negative control. Data (left) represent mean ± SEM of three independent experiments (*n* = 6 seedlings per genotype/treatment in each experiment). Photographs of the seedlings (right) are from one representative experiment. **b** Quantitative reverse transcriptase–PCR (qRT-PCR) analysis of gene expression affected by *lux-1* upon *Pma*DG3 infection (*n* = 2). **c** qRT-PCR analysis of gene expression affected by *eds1-2* upon *Pma*DG3 infection (*n* = 2). **d** Disease rating of *Botrytis*-infected leaves. Plants were sprayed with *Botrytis* (2 × 10^5 spores/ml) 25 or 37 h after onset of LL and the fourth to sixth leaves of each genotype were scored for disease symptoms 4 dpi (121 or 133 h after onset of LL) using the following rating scale: 1 = no lesion or small rare lesions; 2 = lesions on 10–30% of a leaf; 3 = lesions on 30–50% of a leaf; 4 = lesions on 50–70% of a leaf; 5 = lesions on >70% of a leaf. Data (left) represent mean ± SEM of three independent experiments (*n* = 30 leaves per genotype in each experiment). Statistical analysis was performed with one-way analysis of variance with post hoc Tukey honestly significant difference test. For **a–c**, different letters indicate significant difference among the samples at the same time point. For **d**, asterisks indicate significant difference in the same disease scale category between Col-0 and other genotypes. The experiments in **b**, **c** were repeated two times and similar results were obtained

gene expression in LD that, at least for the *CAB2:LUC* or *GRP7:LUC* reporters, is indistinguishable from that in WT seedlings[6]. We confirmed this rhythmic gene expression in *lux-1* in LD by qRT-PCR (Supplementary Fig. 6). We found that *PRR9*, *PRR7*, *PRR5*, and *LUX* showed distinct expression peaks in Col-0, which

are similar to those in *lux-1*. Expression of these genes was higher in *lux-1* than in Col-0 at each time point tested, consistent with LUX-repressing expression of these genes. In addition, we previously showed that *acd6-1* does not affect clock activity[35]. Therefore, we believe that the altered expression of cycling genes

affected by *lux-1* in the RNA-seq analysis is unlikely due to altered circadian phase among Col-0, *lux-1*, *acd6-1*, and *acd6-1lux-1*. Nevertheless, because there were only two time points (ZT1 and ZT13) used in the RNA-seq analysis, we may have missed some cycling genes that are affected by *LUX* at other times of day.

LUX was shown to bind to the LBS motifs in the promoters of *GI*, *LNK1*, *PRR7*, *PRR9*, and *LUX* itself[7,12,51]. We confirmed the binding of LUX to these gene promoters by ChIP experiments (Supplementary Table 1). Further ChIP experiments with selected *LUX*-affected clock genes revealed additional LUX-binding targets, including a second LBS in the *LNK1* promoter (Fig. 3d, the LNK1 position and ref. [51]) and sites in the promoters of *LNK2* (a homolog of *LNK1*[52,53]) and *CDF1* (a flowering repressor gene[48]) (Fig. 3e, f). Overall, our data expand the set of LUX direct targets that participate in clock TTFLs and output pathways.

**Reciprocal regulation of clock activity by JA signaling.** Data from this report and previous studies[16,36,37] have established a circadian control of JA signaling. Whether JA signaling affects clock activity has not been reported previously. To test this possibility, we first examined the expression of several clock genes in Col-0 seedlings treated with 100 μM MJ in LL and found a suppression of *LUX*, *CCA1*, and *GRP7* transcripts within a 24-h treatment (Supplementary Fig. 7).

To further test whether MJ affects clock activity, we monitored luciferase reporter activity driven by the *CCA1* promoter (*CCA1:LUC*) in Col-0 seedlings. MJ inhibited *CCA1:LUC*/Col-0 seedling growth in a dosage-dependent manner (Supplementary Fig. 8a). After normalization of the amplitude to seedling leaf area, we found that MJ dampened the amplitude of *CCA1:LUC*, regardless whether MJ was applied at LL25 or LL37 (Supplementary Fig. 9a). Neither the period nor the phase of *CCA1:LUC*/Col-0 were affected (Supplementary Fig. 9c, d). We further introduced the *CCA1:LUC* reporter into the *coronatine insensitive 1-17* (*coi1-17*) mutant that has an impaired JA receptor[33,34,54]. MJ did not affect seedling growth (Supplementary Fig. 8a) or the amplitude, period, or phase of *CCA1:LUC* rhythmic expression in *coi1-17* (Supplementary Fig. 9e–h). Thus MJ-induced amplitude dampening of the circadian clock requires an intact JA receptor.

MJ suppression of plant growth makes it difficult to distinguish the direct effect of MJ on clock activity from secondary effects due to its growth inhibition. To further test whether JA signaling could reciprocally affect clock activity, we used JA-isoleucine (JA-Ile), a major bioactive JA derivative that binds to COI1 to activate JA signaling[55]. JA-Ile did not cause seedling growth inhibition (Supplementary Fig. 8b), suggesting that these two chemicals act differently to regulate plant growth. Similar to MJ, JA-Ile induced drastic amplitude dampening in both *CCA1:LUC* and *GRP7:LUC* reporters in Col-0 in a dosage-dependent manner (Fig. 5a, b, i, j). The period of both reporters in Col-0 was significantly lengthened, albeit by <1 h, with 100 μM JA-Ile treatment (Fig. 5c, k). We did not observe phase change of the two reporters with JA-Ile treatment (Fig. 5d, l). In addition, JA-Ile did not induce changes of amplitude, period, and phase of *CCA1:LUC* in *coi1-17* (Fig. 5e–h). Thus these data support responsiveness of the circadian clock to JA signaling and, in whole, demonstrate reciprocal regulation between the circadian clock and JA signaling.

## Discussion

Recent studies have established the role of the circadian clock in regulating plant innate immunity. However, the mechanisms underlying this role of the circadian clock are still not well understood. Here we illuminate that the core clock component LUX affects temporal behavior of the stomata to pose physical barrier and modulates key defense signaling mediated by SA and JA, leading to broad-spectrum disease resistance to pathogens and pests. Our data also identify additional *LUX* targets participating in clock TTFLs and output pathways. We further find that activation of JA signaling can feed back to influence clock activity.

Plants are known to employ different mechanisms to fight against pathogens and pests with different lifestyles at different times of day. For infiltrated *P. syringae*, plants with a normal circadian clock show higher susceptibility at night than in the morning (Fig. 2c and refs. [35,37,38]). For epiphytic bacterial pathogens, such as spray-infected *P. syringae*, plants show higher susceptibility in the morning than at night. Because epiphytic bacteria need to pass through stomata to gain access to the interior of plant tissue and the infiltrated bacteria bypass this physical barrier, the differential resistance of plants to pathogens with different infection modes suggests that stomata-independent defense is strong during the day while stomata-dependent defense is dominant at night[35]. Our data show that the *lux* mutants lose the temporal defense demonstrated by Col-0 in response to both infiltrated and sprayed *P. syringae* (Fig. 2b, c), suggesting a role of LUX-mediated circadian control of stomata-dependent and stomata-independent defense.

The *lux* mutants, like Col-0 plants, showed higher resistance to sprayed *P. syringae* in the subjective evening than in the subjective morning, although at both infection time points, the *lux* mutants were more susceptible than Col-0 (Fig. 2b). These observations appear to suggest that the circadian clock does not contribute to defense against epiphytic bacteria. However, our further analysis of the change of stomatal aperture led us to reject this notion. We found that, in LL without *P. syringae* infection, the *lux* mutant lost temporal oscillation of stomatal aperture exhibited by Col-0 (Fig. 2d and Supplementary Fig. 2b). In the presence of *P. syringae*, stomata of Col-0 were highly sensitive for aperture reduction in the morning but showed no response at night due to the closure of stomata. In contrast, the *lux* mutants lost this temporal gating of the response to acute *P. syringae* infection, showing stomatal aperture reduction both in the morning and at night and being even more sensitive to the bacteria at night. Therefore, our data support that the *lux* mutations disrupt the circadian clock and subsequently abolish this temporal variation in stomata-dependent defense. Because the *lux* mutations do not completely abolish *P. syringae*-induced stomatal aperture reduction in the morning, LUX likely only partially affects stomata-dependent defense, and additional factors also contribute to this defense. In contrast, plants use different defense mechanisms in response to challenge by the necrotrophic fungal pathogen *Botrytis* and are less dependent on stomata-dependent defense. Accordingly, the *lux* mutants infected with *Botrytis* in the subjective evening are not more resistant than those infected in the subjective morning (Fig. 4d).

In addition to using physical barriers like stomata to exclude pathogens, plants can mount defense through activating cellular signaling pathways. SA and JA are important defense signaling molecules. SA is generally considered to be important for defense against biotrophic pathogens, whereas JA promotes resistance to necrotrophic pathogens and insects[56–58]. Crosstalk between SA and JA determines the balance of plant defense against pathogens and pests with different lifestyles. However, how the circadian clock affects SA- and JA-mediated defense is complex and not completely understood. We report here a role of *LUX* in regulating SA and JA signaling. The *lux-1* mutation suppresses high SA accumulation in *acd6-1* and results in reduced SA levels during acute defense responses in the presence of *P. syringae* (Fig. 1). RNA-seq analysis reveals that expression of many

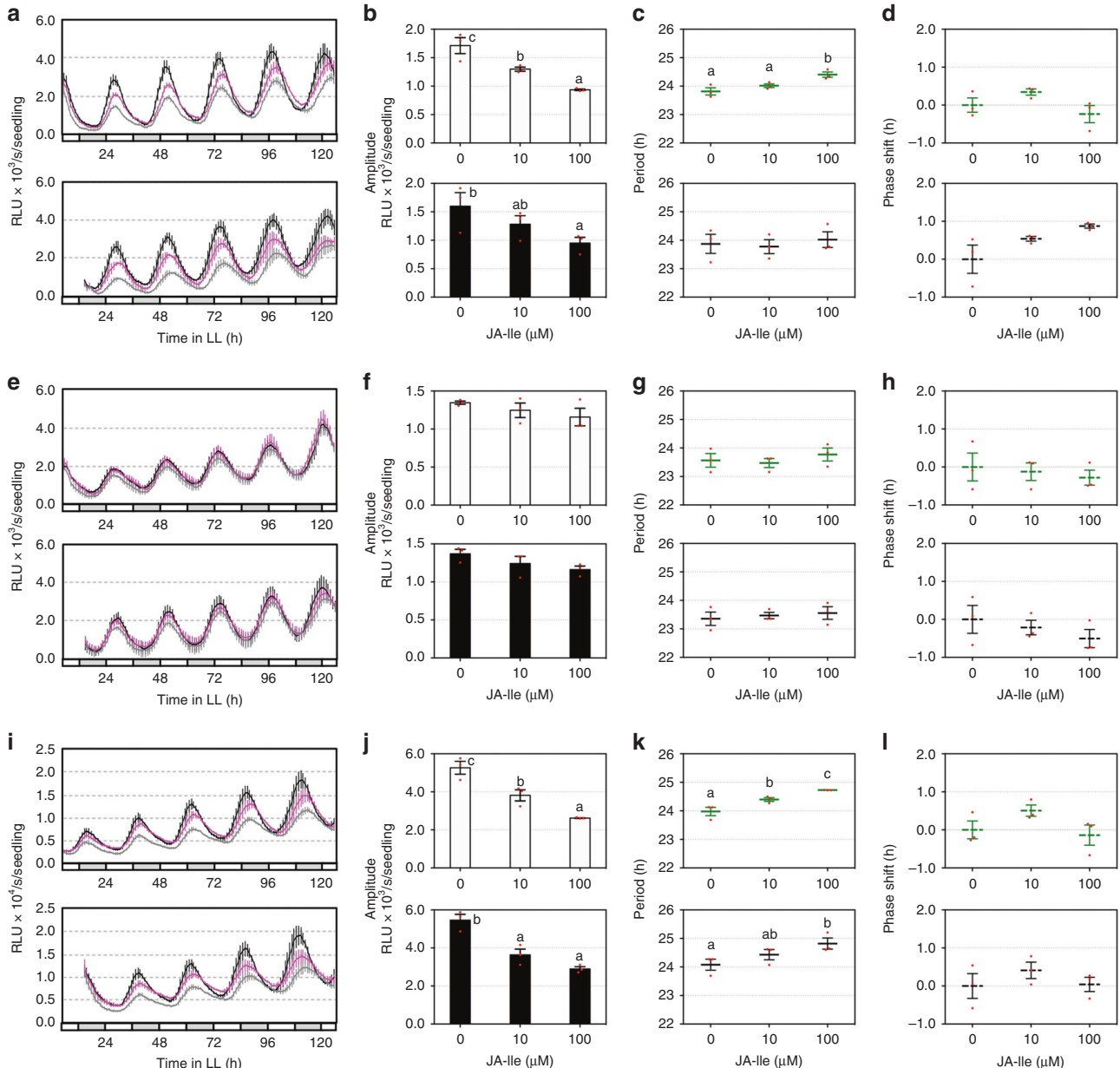

**Fig. 5** Activation of jasmonic acid (JA) signaling reciprocally affects clock activity. Five-day-old seedlings entrained in LD were transferred to LL for 1 day and were treated with JA-isoleucine (JA-Ile). Luminescence was recorded at 1-h intervals for 5 days and analyzed for amplitude, period, and phase with the R package MetaCycle. **a–d** Expression of *CCA1:LUC* in Col-0 treated with JA-Ile 25 h (top) or 37 h (bottom) after onset of LL. **e–h** Expression of *CCA1:LUC* in *coi1-17* treated with JA-Ile 25 h (top) or 37 h (bottom) after onset of LL. **i–l** Expression of *GRP7:LUC* in Col-0 treated with JA-Ile 25 h (top) or 37 h (bottom) after onset of LL. **a**, **e**, **i** Luminescence traces. RLU relative luminescence units. The color indicates JA-Ile concentration, black for 0 μM, magenta for 10 μM, and gray for 100 μM. **b**, **f**, **j** Amplitude. **c**, **g**, **k** Period. **d**, **h**, **l** Phase shift. Data represent mean (±SEM) of three independent experiments (*n* = 8 or 12 for each experiment). Statistical analysis was performed by one-way analysis of variance post hoc Tukey honestly significant difference test. Different letters indicate significant difference among the samples ($P < 0.05$)

SA-related genes are affected by *lux-1*. Our detection of a direct binding of LUX to the promoter of *EDS1*, a major SA regulator involved in the SA signal amplification loop[27,42], suggests that this SA regulatory role of LUX could be, at least in part, mediated through *EDS1*.

The *lux-1* mutant also demonstrates lower sensitivity to MJ treatment and altered JA gene expression relative to Col-0 (Figs. 3a and 4). LUX was previously shown to bind to the promoter of *MYC2*, a JA transcription factor[36]. We report here a direct binding of LUX to the promoter of the JA signaling repressor *JAZ5*. Interestingly, our data show that *eds1-2* is less sensitive to MJ and has increased expression of JA-related genes

upon *P. syringae* infection. Thus *EDS1* is also involved in JA signaling. This role of *EDS1* could be due to *EDS1*-mediated crosstalk between SA and JA signaling. Alternatively, it is also possible that *EDS1* exerts direct influence on JA signaling in an SA-independent manner[27,42]. Indeed, a recent study showed that two EDS1 interactors, PHYTOALEXIN DEFICIENT 4 (PAD4) and SENESCENCE-ASSOCIATED GENE 101[59,60], interacted with MYC2 in a transient assay[61]. EDS1–PAD4 binding could compete for PAD4–MYC2 interaction and therefore affect JA signaling. Together, these data indicate that *LUX* could gate JA signaling by directly controlling multiple JA signaling components.

Our study revealed the role of LUX in mediating SA and JA signaling, thus providing a mechanistic explanation for the crosstalk between the circadian clock and defense. Consistent with this role of *LUX*, loss of function in *LUX* confers compromised resistance to a broad spectrum of pathogens and pests with different lifestyles, including the herbivorous insect *Trichoplusia ni* (cabbage looper), the biotrophic bacterial pathogen *P. syringae*, and the necrotrophic fungal pathogen *Botrytis* (Figs. 2 and 4d)[16]. In addition to LUX, other clock genes may affect plant defense through SA and JA signaling. Among previously reported clock genes with roles in defense regulation, *CCA1* and *LHY* act largely in an SA-independent manner[35]. Like *lux*, arrhythmia caused by overexpression of *CCA1* blocked resistance to cabbage loopers[35,62]. Whether *CCA1* acts similarly as *LUX* in affecting JA signaling and subsequently insect resistance remains to be determined. Another clock gene, *CCA1 HIKING EXPEDITION* (*CHE*), encodes a protein that binds to the promoter of the major SA synthase gene *ICS1*[17]. *CHE* was shown to be important for diurnal SA biosynthesis both in the absence of pathogens and during *P. syringae*-induced SAR. Whether *CHE* gates local acute SA biosynthesis upon pathogen infection is currently unknown. The clock protein TIC was shown to regulate JA signaling through a direct interaction with the JA transcription factor MYC2[37]. It is not known, however, whether TIC acts through the JA pathway to modulate the disease outcome of plants in the presence of pathogens and pests. Thus it would be pertinent to reveal the molecular mechanism underlying the function of clock genes in their control of plant defense.

Because SA and JA are important defense signaling molecules, multiple regulatory inputs to SA and JA signaling from the circadian clock allow plants to continuously monitor the change of these signaling pathways to ensure proper growth, development, and response to external stimuli. Both SA and JA levels as well as expression of some genes involved in SA and JA biosynthesis and signaling oscillate under non-challenged conditions[1]. The cycling expression of most clock genes necessitates the use of clock genes expressed at distinct circadian phases in order to provide inputs to SA and JA signaling at multiple times of day. The circadian clock is also likely important to gate SA and JA signaling under acute stress conditions.

While establishing regulation by LUX of plant innate immunity, we realize the complexity of host–pathogen interactions, the outcome of which is likely influenced by multiple other factors in addition to the circadian clock. We observed enhanced disease susceptibility of the *lux* mutants in LL with a light intensity of 10 $\mu$mol m$^{-2}$ s$^{-1}$ photon flux but not with 180 $\mu$mol m$^{-2}$ s$^{-1}$ photon flux. These results suggest that the defense role of the circadian clock is conditional and influenced by light. The EC, consisting of LUX-ELF3-ELF4, is known to regulate light signaling through affecting expression of many photosynthesis genes and light responsive genes[63]. Thus mutations in *LUX* or other EC genes could make plants particularly sensitive to light, complicating pathogen response. Although 10 $\mu$mol m$^{-2}$ s$^{-1}$ photon flux is a relatively low light intensity, compared with the conditions typically used for plant growth in the laboratory, such light intensities are encountered in deeply shaded conditions and every day during twilight after dawn and prior to dusk. In addition, both LL and DD have been routinely used as free running conditions to test clock activities in plants, animals, and fungi in laboratory conditions. Therefore, our use of this low light regime is physiologically relevant.

In addition to light intensity, other factors, such as light duration and temperature, contribute to LUX-regulated processes. For instance, the early flowering phenotype conferred by the *lux* mutations is more evident in 8 h L/16 h D than in 16 h L/8 h D[6]. The transcriptional targets of LUX (and its interactor

ELF3) are temperature dependent, suggesting a temperature input to EC function[63,64]. Together, these observations suggest the complexity of circadian regulation of biological processes, which can be further compounded by additional factors that modulate the process either directly or indirectly via an effect on the circadian clock.

In addition to defense control, our data support the importance of *LUX* in maintaining clock function, likely through a direct control of expression of core clock TTFL genes and genes in output pathways. A recent ChIP-seq study reported >800 LBSs in Arabidopsis[63], supporting this notion. Our bioinformatics analysis followed by ChIP experiments revealed additional new targets of LUX, including *EDS1*, *JAZ5*, *LNK2*, and a second LBS motif in the *LNK1* promoter. We also detected LUX binding to the *CDF1* promoter, which was shown as one of the LUX targets in the ChIP-seq experiment but had not been independently verified[63]. LUX likely acts as a transcriptional repressor to affect the expression of many target genes. It is also possible that LUX positively regulates gene expression (Figs. 3 and 4b). This gene activation role of *LUX* could be indirect and reflect LUX's repression of another repressor important for the regulation of gene transcription. Alternatively, the LUX protein might activate target gene expression through recruitment to promoters, including that of *EDS1*, as part of as yet undescribed transcriptional activation complex. Together, the detection of widespread targets of LUX in Arabidopsis genome supports the importance of LUX in regulating the circadian clock and other biological processes.

While the circadian clock regulates multiple output pathways, including plant development and responses to environmental stimuli, many of these output pathways are known to reciprocally regulate clock activity[1]. Such reciprocal regulation likely represents a mechanism to integrate environmental cues with proper growth and development. This report and those from other researchers establish a reciprocal regulation of the circadian clock and defense signaling mediated by SA and JA. SA was shown to delay the phase and dampen the amplitude of some clock reporters[65]. However, SA does not affect clock period[35,53,65,66]. We show here that JA-Ile dampens the amplitude and lengthens period of two clock reporters in wild-type Col-0. These observations were largely corroborated by those obtained with an JA-Ile analog, MJ. However, we also observed differences in the change of period and seedling growth with the two chemicals, suggesting that they have overlapping yet also distinct function in signal activation in plants. Together, this reciprocal regulation of the circadian clock by SA and JA signaling provides another layer of monitoring of these defense pathways, which can be reset by their own feedback modulation of circadian clock function. That LUX regulates JA signaling and that *LUX* expression is also influenced by JA clearly suggest LUX is a key, although not necessarily the sole, node in mediating crosstalk between the circadian clock and defense signaling involving JA.

A better understanding of the role of the circadian clock in regulating plant growth, development, and responses to abiotic and biotic stresses in plants would tremendously advance our knowledge of basic science, as well as offer potential applications to improve crop yield and disease resistance. Misexpression of several core clock genes has been shown to compromise plant immunity to pathogens and pests. However, it remains unclear how individual clock genes work together to integrate endogenous and exogenous cues to regulate the circadian clock and defense. We demonstrate in this report that LUX directly targets important clock TTFL genes and output genes for transcriptional regulation and that LUX-mediated defense signaling also reciprocally regulates clock activity. Thus these data provide a mechanistic view of *LUX* function as a pivotal node connecting

the circadian clock and defense. Future work revealing mechanisms of action of additional clock TTFL genes will shed more light on this exciting research field.

## Methods

**Plant materials.** All plants used in this report are in the Col-0 background. Unless otherwise indicated, plants were grown in growth chambers with a light intensity at 180 μmol m$^{-2}$ s$^{-1}$, 60% humidity, and 22 °C either in a 12 h light/12 h dark (LD) cycle. The *lux-1* and *elf3-7* mutants were kindly provided by Todd Michael (Craig Venter Institute); the *lux-4*, *lux-4* expressing *LUX-GFP*, and *CAB2:LUC* seeds were provided by Steve Kay (The University of Southern California); and the *coi1-17* seed was provided by Barbara Kunkel (Washington University). The *CCA1:LUC* reporter was introduced into *coi1-17* by crossing Col-0 expressing *CCA1:LUC* with *coi1-17*, selfing, and identifying F2 lines homozygous for both the *coi1-17* mutation and the *CCA1:LUC* reporter. The *acd6-1lux-1* double mutant was generated by crossing two single mutants and selecting the homozygous double mutant in the F$_2$ generation, using derived cleaved amplified polymorphic sequence markers specific for each mutation. Primers for mutant detections are listed in Supplementary Table 2.

**Pathogen infection.** *P. syringae* pv. *maculicola* ES4326 strains DG3 (PmaDG3), *Pma avrRpm1*, *Pma avrRpt2*, and *Pma HrcC⁻* were used in this report. Freshly cultured bacteria were resuspended in 10 mM MgSO$_4$. Plants were infiltrated or evenly sprayed with a bacterial solution at the indicated concentration, time, and light condition.

For SAR induction, the fourth to sixth leaves of 25-day-old plants grown in LD were infiltrated with *Pma avrRpt2* (optical density (OD) = 0.05). The leaves with the primary infection were mostly dead (a hypersensitive response) at 2 dpi and were detached. The adjacent two to three leaves were further infiltrated with *Pma*DG3 (OD = 0.0001) and the bacterial growth was assessed at 3 days post the secondary infection. Because the hypersensitive response in Arabidopsis was reported to be light dependent[67], both the primary and secondary infections were conducted in LD.

*B. cinerea* strain BO5-10 was kindly provided by Tesfaye Mengiste (Purdue University). Twenty-five-day-old plants grown in LD were moved to LL for 1 day before being sprayed with *Botrytis* (2 × 10$^5$ spores/ml) 25 or 37 h after onset of LL. Plants were further transferred 24 hpi to low-light LL with a light intensity of 10 μmol m$^{-2}$ s$^{-1}$. Disease symptoms of leaves at the fourth to sixth positions were rated at 4 dpi (121 or 133 h after onset of LL). The rating scale was as the follows: 1 = no lesion or small rare lesions; 2 = lesions on 10–30% of a leaf; 3 = lesions on 30–50% of a leaf; 4 = lesions on 50–70% of a leaf; 5 = lesions on >70% of a leaf.

**SA quantification.** Leaves from *P. syringae*-infected or mock-treated 25-day-old plants or non-infected whole plants were collected at the indicated times for SA extraction and measurement by a high-performance liquid chromatography instrument (Shimadzu LC-20AT).

**Cell death staining.** The fifth to seventh leaves of 25-day-old plants were stained with a trypan blue solution for cell death and photographed with a Leica IC80 HD camera connected to a Leica M80 stereomicroscope.

**RNA analyses.** Total RNA was extracted with TRIzol reagent (Invitrogen) according to the manufacturer's instruction. Primers used in qRT-PCR are listed in Supplementary Table 2.

**Stomatal aperture measurement.** Twenty-five-day-old plants grown in LD (180 μmol m$^{-2}$ s$^{-1}$ light intensity, 60% humidity, 22 °C, and 12 h light/12 h dark cycle) were moved to LL (10 μmol m$^{-2}$ s$^{-1}$ light intensity, other parameters remained the same) for 1 day. The fifth to seventh leaves of each genotype were taken at LL25 or LL37 and mounted onto scotch tape at the abaxial side. The top layer of a leaf was peeled off and the tape with the lower layer of the leaf was cut and mounted on a glass slide for observation with an inverted microscope. For *P. syringae* treatment, leaves were incubated with *Pma*DG3 (OD$_{600}$ = 0.1) or sterile water at LL25 or LL37. Leaves were processed as above for stomata imaging at 0, 1, and 3 hpi. Immediately after leaf processing, images of at least three random regions of each of the three or more leaves of each genotype per treatment were taken with a camera (Canon Digital Rebel xsi, Japan) connected to an inverted microscope (Olympus Model IMT-2). Each stoma was measured for the width and the length using ImageJ (version 1.45). Stomatal aperture was determined by the ratio between the width and the length of a stoma. At least 80 stomata were used for calculating the average stomatal aperture of each genotype per treatment.

**Seedling growth assay with flg22 or MJ treatment.** Seeds were surface-sterilized in a bell jar with bleach vapor generated by slowly adding 3 ml of concentrated HCl into 100 ml bleach. The jar was immediately sealed with a lid for 3 h in a fume hood. Sterilized seeds were plated on agar plates containing 1/2 MS media supplemented with 1% sucrose (pH 5.7). The plates were incubated at 4 °C for 2 days

before being transferred to a tissue culture chamber in LD and 22 °C for 4 days. Seedlings were then transferred to a 24-well tissue culture plate with sterile water in the presence or absence of 1 μM flg22 or 10 μM MJ. At least four seedlings were used for each genotype per treatment. Root length of the seedlings was measured 4 days later. The Ws ecotype that does not have a functional flg22-receptor FLS2[41] and the *jar1-1* mutant that is insensitive to JA signaling[68] were used as a negative control for flg22 and MJ treatments, respectively.

**Callose staining.** Flg22- or *Pma HrcC⁻*-treated leaves were harvested at 24 hpi and boiled in alcoholic lactophenol (95% ethanol:lactophenol = 2:1) for 2 min followed by rinsing in 50% ethanol. Aniline blue solution (0.01% aniline blue in 150 mM KH$_2$PO$_4$, pH 9.5) was used to stain the leaves for 1.5 h in dark. At least four leaves of each genotype were used in each treatment. Callose deposition was visualized with a Leica fluorescence stereomicroscope (M205 FA) and imaged with a CCD camera (Cool Snap HQ$^2$, Photometrics, USA). Callose deposits were quantified using ImageJ (version 1.45).

**RNA-seq analysis.** Total RNA was extracted from 25-day-old Col-0, *lux-1*, *acd6-1*, or *acd6-1lux-1* plants collected at ZT1 or ZT13 1 and 13 h after light onset, respectively. Triplicate biological samples were used for most genotypes at each time point, except *acd6-1* and *acd6-1lux-1* at ZT13, which had duplicate samples. In all, 0.5 μg RNA per replicate was used to generate cDNA libraries using the Illumina TruSeq RNA Sample Preparation Kit (catalog no. RS-122-2001). The samples were multiplexed and sequenced using the Illumina HiSeq sequencing platform in Genomics Resources Core Facility at Weill Cornell Medical College. Sequencing was conducted with a standard run of 51 cycles and single reads[31]. At least 150 million reads per lane were obtained for sequencing. Differentially expressed genes in each comparison group were identified using the R package DESeq, using the default parameters[69]. The default false discovery rate of 0.1, which results in statistical significance with *P* values < 0.001, was used to define significant difference in the gene expression in each comparison group.

For global gene expression profiling, the relative expression value (reads per kilobase of transcript per million mapped reads (RPKM)) of >0.3 was used as the cutoff to include genes for further analyses. Each RPKM value was corrected by adding the number one and then was log$_2$-transformed for generating the correlation dendrogram with R function cor(). To show the number of overlapped genes affected among the comparison groups, Venn diagrams were generated with the R package VennDiagram. The GO annotations tool on the TAIR website was used to assign genes into functional groups. The whole-genome genes of Arabidopsis were also similarly analyzed to provide the reference number for GO annotation. Graphical representation of gene expression correlation was produced by the heatmap.2() function in the R package gplots.

**ChIP assays.** The ChIP experiments were conducted according to previous descriptions with modifications[7,70]. Fourteen-day-old wild-type *CAB2:LUC* or *LUX-GFP* (*lux-4* expressing *LUX:LUX-GFP*) seedlings grown on soil were harvested at ZT13. The seedlings were crosslinked by immersion in 30 ml of 1% formaldehyde solution at room temperature under vacuum for 20 min. Glycine (2 ml at 2 M concentration) was added to the seedlings for 5 min under vacuum to quench the crosslink reaction. The seedlings were then rinsed with water and stored at −80 °C for further use. The seedlings were ground in liquid nitrogen to a fine powder and extracted for DNA and protein with 30 ml of extraction buffer I (0.4% sucrose, 10 mM Tris-HCl pH8.0, 0.035% 2-mercaptoethanol, 1 mM phenylmethanesulfonylfluoride (PMSF), 5 mM benzamidine, 1× Roche protease inhibitors) in a 50 ml Falcon tube. The tube was incubated on ice for 15 min until the tissue was thawed. The solution was filtered through miracloth and centrifuged at 750 × *g* at 4 °C for 20 min. The pellet was resuspended in 1 ml of extraction buffer II (0.25 M sucrose, 10 mM Tris-HCl pH8.0, 10 mM MgCl$_2$, 1% Triton X-100, 0.035% 2-mercaptoethanol, 1 mM PMSF, 5 mM benzamidine, 50 μM MG132, 1× Roche protease inhibitors) and centrifuged at 15600 g at 4 °C for 10 min. The remaining pellet was resuspended in 500 μl of extraction buffer III (1.7 M sucrose, 10 mM Tris-HCl pH8.0, 2 mM MgCl$_2$, 0.15% Triton X-100, 0.035% 2-mercaptoethanol, 1 mM PMSF, 5 mM benzamidine, 50 μM MG132, 1× Roche protease inhibitors). The suspension was laid on the top of another 500 μl extraction buffer III in a 1.5 ml Eppendorf tube and centrifuged at 15,600 × *g* at 4 °C for 1 h. After removing the supernatant, the pellet was resuspended in 500 μl of nuclei lysis buffer (50 mM Tris-HCl pH8.0, 10 mM EDTA, 1% sodium dodecyl sulfate (SDS), 1 mM PMSF, 5 mM benzamidine, 50 μM MG132, 1× Roche protease inhibitors). The suspension was sonicated on ice (15 s pulse followed by 45 s resting, repeating 12 times) using a sonicator (Virsonic Cell Disruptor, Model 16-850) set at 30% power to shear genomic DNA to an average size of 500–1000 bp. The sonicated chromatin solution was centrifuged at 15,600 × *g* for 10 min at 4 °C and the supernatant was collected. This process was repeated one time and all supernatants containing the chromatin solution were combined. One hundred and fifty microliters of chromatin solution was then diluted in 1350 μl of ChIP dilution buffer (16.7 mM Tris-HCl pH8.0, 1.2 mM EDTA, 1.1% Triton X-100, 167 mM NaCl, 1 mM PMSF, 5 mM benzamidine, 50 μM MG132, 1× Roche protease inhibitors). Anti-GFP antibodies (IP) (Abcam, product code 290) or whole rabbit IgG (mock) (Jackson ImmunoResearch cat 011-000-003) coated onto Dynabeads

Protein G (Invitrogen cat 100.04D) were used for IP. Diluted chromatin solution (700 μl/sample) was incubated with either IP- or mock-treated beads at 4 °C for 1.5 h. In the meantime, 70 μl diluted chromatin solution was kept aside as the input. After incubation, beads were washed at 4 °C with low salt buffer twice (20 mM Tris-HCl pH8.0, 150 mM NaCl, 0.2% SDS, 0.5% Triton X-100, 2 mM EDTA), high salt buffer once (20 mM Tris-HCl pH8.0, 500 mM NaCl, 0.2% SDS, 0.5% Triton X-100, 2 mM EDTA), and TE buffer twice. The beads were resuspended in 100 μl elution buffer (50 mM Tris pH8.0, 10 mM EDTA, 1% SDS) and incubated at 65 °C for 15 min followed by 30 s centrifugation at $376 \times g$ to elute the immunocomplexes from the beads. The supernatant was transferred into a new 1.5 ml Eppendorf tube and was added with NaCl to the final concentration of 0.2 M and 1 μl of Proteinase K. All tubes (IP, mock, and input tubes for each sample) were incubated overnight at 65 °C to reverse crosslinking. Chromatin DNA was purified using a Qiagen QIAquick PCR Purification Kit and resuspended in 300 μl of water. ChIP DNAs were quantified by real-time qPCR with primers specific for the amplicons covering LBS motifs in gene promoters using a 3 μl aliquot from IP, mock, or input tubes for each sample. Fold enrichment for each promoter region in plants expressing *LUX: LUX-GFP* or *CAB2:LUC* was normalized with the input DNA and the internal control DNA (a fragment from the *UBQ* gene promoter) and was calculated using the following equation: $2^{(Ct\ input\ -\ Ct\ IP)}/2^{(Ct\ input\ -\ Ct\ mock)}$. Primers for the ChIP experiments are listed in Supplementary Table 2.

**Luciferase assay.** Seedlings expressing the *CCA1:LUC* or *GRP7:LUC* reporter were grown on 1/2 MS media with 1% sucrose in LD and at 22 °C for 5 days. Seedlings were transferred to 96-well plates containing 200 μl of 1/2 MS medium with 0.5% sucrose, 0.4% agar, and 0.25 mM D-luciferin for 1 day in LD followed by 1 day in LL with a light intensity of 180 μmol m$^{-2}$ s$^{-1}$. Each well contained one seedling. Seedling treatments were conducted at LL25 or LL37 by adding 15 μl of each chemical (MJ (10 μM or 100 μM), JA-Ile (10 μM or 100 μM), or the mock solution (sterile water)) to each well. Immediately after the treatments, the plants were measured for luminescence with an Omega Luminescence Reader (BMG LAB-TECH, Inc.) in LL with 90 μmol m$^{-2}$ s$^{-1}$ photon flux density. LUC activity was measured at 1-h intervals for 5 days. Each microplate with seedlings was photographed after LUC recording. Leaf area of each seedling was measured using ImageJ (version 1.45). The amplitude, period, and phase were calculated with the R package MetaCycle[71].

**Reporting summary.** Further information on research design is available in the Nature Research Reporting Summary linked to this article.

## Data availability

Source data are provided in the paper, its supplementary files, and a Source Data file for Figs. 1b, d–g, 2b–g, 3b–f, 4a–d, 5b–d, f–h, j–l and Supplementary Figs. 1, 2, 5, 6, 7, 8 and 9. The RNA-seq data are deposited in the National Center for Biotechnology Information Gene Expression database with accession number GSE115680.

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

## Acknowledgements

We thank the members in the Lu laboratory for their assistance in this work. We thank Dr. Dorothee Staiger and Dr. Jane Parker for helpful discussions about this report. We thank Dr. Hua Jian, Dr. Yuelin Zhang, and Dr. Jose Pruneda-Paz for sharing with us the ChIP protocols. This work was supported by a grant from National Science Foundation (NSF 1456140) to H.L. and C.R.M.

## Author contributions

C.Z. and M.G. did most experiments with pathogen infection, qRT-PCR, and clock assays; M.G. and R.E. did stomata measurement. N.C.S. characterized *lux-1* in *acd6-1* background; W.A., A.H., T.D. and D.L. assisted in *lux-1* characterization; L.W. assisted in clock assays; O.D. and N.A. helped with bioinformatics analysis; X.W. and C.R.M. provided valuable research input; H.L. designed experiments, participated in some experiments, and wrote the manuscript with help from C.Z., M.G. and C.R.M.

## Additional information

**Competing interests:** The authors declare no competing interests.

