## [Peer Review File · Nature Communications]

Reviewers' comments:

Reviewer #1 (Remarks to the Author):

The manuscript by Zhang et al. first identified a circadian clock component, LUX, as a potential player in SA-mediated defense through genetic complementation assay using *acd6-1*, a mutant with constitutive defense. Then the authors performed a comprehensive plant defense phenotype characterization of the *lux* mutant to prove its role in plant immunity. Combining RNA-seq analysis and ChIP analysis, the authors suggested that LUX may execute its defense function through transcriptional regulation on defense genes. Finally, the authors revealed that coronatine, a JA mimicry, can change the period of circadian clock. In conclusion, the authors claimed that LUX coordinates the circadian clock and plant defense.

While the defense role of LUX has been suggested by Goodspeed et al. previously, the more extensive characterization of LUX can still be useful information to the circadian clock and plant immunity research fields. However, the current form of the manuscript suffers from the following issues.

Major points

1. Based on the title and the abstract, the authors tried to establish LUX as a key gene coordinating the circadian clock and plant defense. While the authors routinely used measurements at ZT1 and ZT13 to account for the role of the circadian clock, the characterization of the defense phenotypes of LUX has not been extensively performed in a circadian fashion, i.e. under free-running condition with at least 4-6 time points. Indeed, according to Figure S2, *lux-1* did not show significant bacterial growth phenotype under free-running condition. Whether other defense phenotypes of *lux-1* observed under LD will still hold under LL is questionable.
2. While the authors claimed that LUX regulates SA-mediated defense, the authors did not show the SAR phenotype of *lux-1*.
3. The section about the coronatine digressed from the major logic flow of the whole manuscript, especially considering that coronatine does not change LUX expression.

Minor points

1. The authors need to provide the p-value cutoff used for DEseq analysis. Did the authors used two-way ANOVA to claim the interaction between *acd6-1* and *lux-1*?
2. The authors need to perform statistical analysis to support the claims in Figure S7.
3. Line 288, EDS1 may regulate JA simply due to the crosstalk between SA and JA signaling .
4. The authors need to deposit their RNA-seq data to public domains for review and re-use of the data.

Reviewer #2 (Remarks to the Author):

This paper proposes that *lux arrhythmico* (LUX) which is part of the evening complex of the Arabidopsis clock, plays a role in coordinating temporal defences in plants. The authors demonstrate this through pathogen assays with *Pseudomonas syringae* and *Botrytis cinerea*, salicylic acid measurements, RNA-seq analyses and ChIP assays. The data support the basic proposal, but there are some points that need clarification and/or correction. Although generally well-written, the manuscript suffers in a few places due to poor grammar and language use; this should be remedied. The findings are novel and do add to our understanding of temporal regulation of defence, as well as our understanding of clock function in plants. Interestingly, the authors propose that LUX can also act as an activator, but do not provide direct evidence for this. This seems to be an overinterpretation of the data. The authors do not comment on the role of other clock components that are affected in the loss-of-function *lux1* mutant. The paper might be enhanced by this analysis, perhaps in place of the EDS experiments which seem a little too

peripheral to the main thrust of the paper?

1. Figure 1. It would be helpful to show the levels of SA in wild-type plants (D) and the bacterial numbers in wild-type plants (F) too. Is the oscillation in SA as seen in the Goodspeed et al. (2012) study detectable in the wild-type (or lux1) – can't tell from G? What time (ZT) did the infection in G take place?
2. Figure 2. What time did infection with *Botrytis cinerea* take place? What strain of *Botrytis*? How was the infection done – detached leaf? This information is not in methods.
3. Figure 6. Please plot data in figures D and E on y-axes with same scales.
4. Figure S2. What do uninfected lux1 plants look like compared to wild-type? Do they start off with less chlorophyll? This is not convincing without this information.
5. Lines 89-104 are inappropriate for the Introduction. The paragraph in lines 109-118 is more suitable.
6. Lines 75-76: this is not true. Goodspeed et al. (2012) did not demonstrate that lux and cca1 affected susceptibility to insects. They used the lux2 mutant and the CCA1-ox line as arrhythmic plants to demonstrate that clock function was responsible for the phase dependent resistance. This should be removed.
7. Line 178: remove 'a' from 'time of a day'
8. Lines 194-198: References should be provided for the expected figures of cycling genes. Furthermore, the way that this is worded makes it difficult to understand what the authors mean. Do they mean that of the genes the current study found to be affected by LUX, 26.7% had previously been demonstrated to cycle under LD conditions, and a further 26.3% to cycle under LL conditions? Or are they saying that of the genes the current study found to be affected by LUX, only 26.7% and 26.3% respectively had previously been demonstrated to cycle under LD or LL?
9. Lines 219-221: The expression of group III genes generally being lower in lux1 does not necessarily equate to LUX having transcription activating activity. The authors need to consider that lack of LUX may lead to the reduction of another repressor, which then results in upregulation of some genes. This assertion is repeated in line 380 and should be treated with caution.
10. Lines 265-267: Goodspeed et al. (2012) did not demonstrate that lux affected susceptibility to cabbage loopers. They used the lux2 mutant as an arrhythmic line (as well as the CCA1-ox arrhythmic line) to demonstrate that clock function was responsible for the phase dependent resistance. This should be corrected in line 373 too.
11. Rephrase to make clearer: lines 509-510 "To determine if a LUX-affected gene cycles during a day, the web-based tool Phaser was used to analyze gene expression under one diurnal..."

Reviewer #3 (Remarks to the Author):

In this manuscript, Zhang et al. describe a role for the Arabidopsis clock transcription factor LUX ARRHYTHMO on the regulation of plant defense responses. Authors indicate that lux-1 mutant plants have compromised disease resistance against *P. syringae* and *botrytis* infections and both SA and JA signaling. Using RNAseq LUX downstream target genes are identified. In particular, LUX was confirmed to bind to the promoters of EDS1 and JAZ5, which are involved in the SA and JA

signaling pathways respectively. Finally, authors found that coronatine (a bacterial chemical that mimics some JA functions) but not MeJA treatments resulted in a longer clock period phenotype.

In the present format the manuscript is too preliminary and needs to be refocused to answer a specific question. If authors are attempting to establish a role for LUX as a coordinator of clock and defense responses, then experiments should be performed in constant conditions. However, it seems that LD was the condition used for most experiments, thus results reflect the role of LUX on defense regulation either by light/dark changes and the clock, or just light/dark changes (if this is the case then the interpretation of most results should be reconsidered). For example, while infections in LD (figure 2A) indicate that *lux-1* plants are more sensitive to *P. syr*, infections in LL do not show any difference suggesting that LUX specific function depends on the presence of LD cycles. In fact, authors observed that, differential susceptibility to ZT1 versus ZT13 infections in LL is identical in wild type and *lux-1* plants. Regarding this later experiment LL "morning" infections should have been performed at ZT25 rather than ZT1 (as only after ZT12 plants are in free running conditions). Thus, overall experiments presented here indicate that LUX mediates defenses (*P. syr* and *botrytis*) in light/dark cycles rather than the clock. Regarding the mechanisms, it seems that LUX regulates callose deposition upon flagellin perception, however results are far less convincing regarding its function on SA production in the absence of the *acd6-1* mutation (Fig. 1G).

Analyses of RNAseq experiments should have been extended to all LUX regulated genes, not specific "lists" of genes. Furthermore, the identification of novel LUX "TTFL" (see comment below) target genes, although interesting, is not relevant for this manuscript. More importantly, follow up experiments on identified genes, such as *EDS1* and *JAZ5*, should be included to establish their role in mediating LUX regulation of defense responses.

Finally authors explored the regulation of Arabidopsis clock function by MeJA and coronatine. I find that these experiments are very preliminary and not conclusive regarding the role of MeJA in clock regulation (and probably for coronatine as well). To properly establish if MeJA does or does not have a role in clock regulation, a series of experiments should have been performed treating plants at different times of the day (this is critical as most clock responses are gated at specific times of the day). If MeJA indeed does not regulate the clock function, what about JA-Ile?. More importantly, given that authors propose that LUX mediates JA responses, does LUX mediate clock responses to coronatine?.

Specific comments:

All text sections require editing.

Additional experimental details should be added both to the materials and methods sections and legends for main figures. (I found the legends for supplementary figures much more informative than those for main figures)

The use of the term "TTFL genes" to refer to clock genes is rather confusing as TTFLs are not exclusive for the clock function.

How were plant sizes determined in Fig. 1B?

Total SA levels reported in Fig 1D for the *acd6-1* mutant are ~10 times higher than what authors published before in Zhang et al. 2013. What would be the reason for such difference?

RNA blots (fig. 1E) should be quantified (or better, PR1 levels determined by Q-PCR). If LUX down regulates SA signaling then PR1 expression should be tested in *col-0* and *lux-1* plants upon SA treatment.

Bacterial titers should be normalized to 1 mm²

Authors state the SA levels oscillate during the day (line 110), however this does not seem to be the case according to the results in figure 1G.

Images in Fig. 2C don't seem representative of the results shown in Fig. 2D (at least for lux-1). To better support these results the comparison between treated and untreated plants for each genotype should be presented.

Callose deposition quantification should be normalized per mm² (Fig. 2F)

Text references to the manuscript by Goospeed et al (2012) should be revised, as this manuscript indicated that the overall clock function (rather than specifically LUX) regulates plant herbivory resistance.

ZT values in figures 6D and 6E are incorrect. ZT or "zeitgeber time" provides a reference to the last dark to light transition.

Regarding luciferase assays:

What was the light intensity in LL?

Plants were grown in LD and then treated with cor or MeJA. Either to perform the treatments or after the treatment these plants had to be moved to a different plate. How did root damage was prevented in this process? I wonder if plant manipulation and/or tissue damage (which could result in JA production) had an effect on clock rhythms. A set of plants that are not treated or manipulated should be processed in parallel to address this potential issue.

It is indicated that seedlings were "briefly" soaked in coronatine or MeJA solutions (the length of treatment should be provided).

The time of day at which treatments were performed should be indicated.

Fig S2A, axis label is missing.

Reviewer #4 (Remarks to the Author):

The authors demonstrated comprehensive studies to understand molecular mechanisms underlying relationship between circadian clock and innate immunity in Arabidopsis. They found that lux1-1 mutants suppressed constitutive defense phenotypes of acd6-1. Phenotypic and genome-wide gene expression analyses of lux-1-1 suggested possible molecular mechanism of LUX for innate immunity. Finally, they showed that strong antagonist of JA, coronatine, can modulate circadian period.

The paper provides potentially interesting results for understanding of molecular link between clock and innate immunity. However, I have concerns about experimental design, data, and interpretation of results.

The authors used only lux-1 allele in this study. While we know that lux-1 and other lux alleles show same circadian phenotypes (e.g., Hazen et al., PNAS 2005), it is still unknown whether immunity-related phenotypes of lux-1 are allele specific (Figure 1, 2). Immunity-related phenotypes of other lux alleles or genetic complementation assay are required to conclude that LUX regulates immunity. I also wonder if elf3 and elf4 show immunity-related phenotypes, because LUX, ELF3, and ELF4 form Evening Complex. RNAseq data of lux-4, elf3 and elf4 mutants might be useful.

The authors claimed that LUX regulates immune responses through EDS1. I felt this is very likely.

However, genetic study using lux1/EDS1 overexpression is useful to consider whether EDS1 has major role in LUX-dependent immune regulation.

Expression changes of LUX-target genes previously reported (e.g., PRR9, PRR7, and GI, from Ezer et al., Nature Plants 2017) in lux-1 seem too little (Figure 3). This figure is not convincing to support that LUX is involved in clock and immune response. I recommend other data presentation styles to say LUX control clock genes, but this was already done by previous studies (Helfer et al., Curr. Biol., 2011, Ezer et al., Nature Plants 2017).

I think the data represented in Figure 6 are too preliminarily and necessary for this manuscript. First, effects of COR was not so strong. Indeed, clock period was lengthened with very high concentration of COR (100 μ M) than that used in immunity papers (low μ M range), suggesting that COR effect on clock is artifact. Light and temperature, two major signals regulating the clock, can alter period length more significantly. How many potent JA analogues were tested, and how many compounds affect the period? Do these compounds really affect only JA signaling in plants? Did knock-down of JA signaling genes alter circadian period? In addition, COR effects on clock TTFL genes were very small (Figure S 8). Further experiments are needed to conclude that COR regulates clock.

minor comments.

We do not use ZT for time value for samples analyzed under constant light conditions. In stead, time in constant light (h) is used.

LUX binds to LNK1 promoter (Mizuno et al., Plant Signal Behav., 9, e28505).

Definition of clock-related genes is poor, though the authors mentioned that these genes were from a list of genes involved in rhythmic processes in TAIR website. Does clock-related genes contain clock-output genes? If so, they contain lots of genes not involved in core clock function.

Responses to Reviewers' Comments

Reviewer #1 (Remarks to the Author):

The manuscript by Zhang et al. first identified a circadian clock component, LUX, as a potential player in SA-mediated defense through genetic complementation assay using *acd6-1*, a mutant with constitutive defense. Then the authors performed a comprehensive plant defense phenotype characterization of the *lux* mutant to prove its role in plant immunity. Combining RNAseq analysis and ChIP analysis, the authors suggested that LUX may execute its defense function through transcriptional regulation on defense genes. Finally, the authors revealed that coronatine, a JA mimicry, can change the period of circadian clock. In conclusion, the authors claimed that LUX coordinates the circadian clock and plant defense.

While the defense role of LUX has been suggested by Goodspeed et al. previously, the more extensive characterization of LUX can still be useful information to the circadian clock and plant immunity research fields. However, the current form of the manuscript suffers from the following issues.

Major points

1. Based on the title and the abstract, the authors tried to establish LUX as a key gene coordinating the circadian clock and plant defense. While the authors routinely used measurements at ZT1 and ZT13 to account for the role of the circadian clock, the characterization of the defense phenotypes of LUX has not been extensively performed in a circadian fashion, i.e. under free-running condition with at least 4-6 time points. Indeed, according to Figure S2, *lux-1* did not show significant bacterial growth phenotype under free-running condition. Whether other defense phenotypes of *lux-1* observed under LD will still hold under LL is questionable.

We have now provided pathogen response data in LL to allow robust conclusions regarding the role of the circadian clock in pathogen sensitivity. To address potential allele specificity, we have used two distinct mutant lines, each homozygous for one of two independently derived *lux* alleles. We also tested a *lux-4* line complemented with LUX-GFP (LUX-GFP), Col-0, and other genotypes. (Figures 2B and 2C). These data support the conclusion that LUX is a positive regulator of resistance against the bacterial pathogen *P. syringae* and the fungal pathogen *Botrytis*.

During the course of our study, we also realized the complexity of host-pathogen interactions, which clearly require a functional circadian clock but can also be influenced by other factors, such as light. We initially tried to address this comment by growing and infecting the plants with the same light intensity as plant growth ($180 \mu\text{mol m}^{-2} \text{s}^{-1}$). As we had reported in the original submission, with this light intensity, we saw enhanced disease susceptibility in the *lux-1* mutant to *P. syringae* spray-infection in LD but not in LL (now Figure S2). Similarly, *Botrytis* symptoms were also much reduced with this light intensity. However, we also tested a lower light intensity ($10 \mu\text{mol m}^{-2} \text{s}^{-1}$) for infection experiments in LL and observed enhanced disease susceptibility to both *P. syringae* and *Botrytis* of loss of function *lux* mutant plants (Figures 2B & 2C), providing critical data to demonstrate the role of LUX-mediated clock in defense

regulation. We believe that studies independent from this report should be conducted to systematically investigate how light (intensity, period, wavelength) affects different defense responses.

2. While the authors claimed that LUX regulates SA-mediated defense, the authors did not show the SAR phenotype of *lux-1*.

We have included the SAR data in this revision (Figure 2D). Both *lux* mutants showed a lack of SAR in our experiments, supporting a role of *LUX* in SAR regulation.

3. The section about the coronatine digressed from the major logic flow of the whole manuscript, especially considering that coronatine does not change LUX expression.

In addition to this reviewer, two other reviewers raised questions regarding our clock assays and the results about the roles of coronatine and MJ in reciprocal regulation of the circadian clock.

Reviewer 3: Finally authors explored the regulation of Arabidopsis clock function by MeJA and coronatine. I find that these experiments are very preliminary and not conclusive regarding the role of MeJA in clock regulation (and probably for coronatine as well). To properly establish if MeJA does or does not have a role in clock regulation, a series of experiments should have been performed treating plants at different times of the day (this is critical as most clock responses are gated at specific times of the day). If MeJA indeed does not regulate the clock function, what about JA-Ile?. More importantly, given that authors propose that LUX mediates JA responses, does LUX mediate clock responses to coronatine?.

Reviewer 4: I think the data represented in Figure 6 are too preliminarily and necessary for this manuscript. First, effects of COR was not so strong. Indeed, clock period was lengthened with very high concentration of COR (100 μ M) than that used in immunity papers (low μ M range), suggesting that COR effect on clock is artifact. Light and temperature, two major signals regulating the clock, can alter period length more significantly. How many potent JA analogues were tested, and how many compounds affect the period? Do these compounds really affect only JA signaling in plants? Did knock-down of JA signaling genes alter circadian period? In addition, COR effects on clock TTFL genes were very small (Figure S8). Further experiments are needed to conclude that COR regulates clock.

We agree with the reviewers that in the original submission, the clock assay was not well designed and the inclusion of coronatine data digressed from the major logic flow of the manuscript. We appreciate these comments, and in response have improved our experimental design. These new experiments support the reciprocal regulation of the circadian clock by JA signaling activation. The major changes we made to address these reviewers' points are listed below:

i) We modified our clock assay by inclusion of 1d in LD and 1d LL for seedlings to adapt to 96-well plates. To avoid variation in the treatment time among individual seedlings when using the dipping method previously described (dipping the seedlings briefly in a solution), we added the chemicals directly to the seedlings in the wells. The detailed protocol can be found in the

Methods under the Luciferase assay subtitle.

ii) We conducted the clock assays with two JA analogs (two doses each of MJ and JA-Ile with morning and evening applications). In addition to Col-0 expressing *CCA1:LUC* or *GRP7:LUC*, we also included the JA-receptor mutant, *coi1-17* expressing *CCA1:LUC*, to provide orthogonal confirmation of JA signaling affecting clock activity. Our data strongly support that activation of JA signaling could reciprocally regulate the circadian clock (Figures 5 and S7).

iii) In the original submission, we reported that MJ has smaller effect than coronatine on clock regulation based on two high throughput gene expression datasets ^{1,2}. However, one experiment was with 5-week-old Arabidopsis Col-0 plants ¹ and the other was with plate-grown seedlings ². The difference in clock gene regulation in the two reports could be due to developmental differences in plant response to these chemicals and may not necessarily support our conclusion. Consistent with this idea, we observed suppression of some clock gene expression with MJ treatment in Arabidopsis seedlings (new data, Figure S6). Thus we did not include the analysis of the two high throughput expression datasets in this report.

iv) We accept the reviewers' reservations about our experiments with coronatine and accordingly have not included the coronation-related data in this revised manuscript.

Minor points

1. The authors need to provide the p-value cutoff used for DEseq analysis. Did the authors used two-way ANOVA to claim the interaction between *acd6-1* and *lux-1*?

We used the R package DESeq with the default parameters ³ to identify differentially expressed genes in each comparison group. The default false discovery rate of 0.1, which results in P-values less than 0.001, was used to define significant difference in gene expression. We clarified this point in the Methods under the RNAseq analysis subtitle.

We did not use two-way ANOVA to claim the interaction between *acd6-1* and *lux-1* although we do see that *acd6-1* likely has much stronger effect than *lux-1* on global gene expression, based on the cluster dendrogram analysis (Figure S3). Our focus of this report is to identify LUX-affected genes. Thus we only compared four groups: a. Col-0 vs. *lux-1* at ZT1; b. Col-0 vs. *lux-1* at ZT13; c. *acd6-1* vs. *acd6-1lux-1* at ZT1; and d. *acd6-1* vs. *acd6-1lux-1* at ZT13. We found that a total of 1618 genes was differentially affected by *lux-1* in at least one of the comparison groups (Table 1, Figures 3 and S4).

2. The authors need to perform statistical analysis to support the claims in Figure S7.

Statistical analyses have been provided to this figure (now Figure S4) and other quantitative figures in this revision.

3. Line 288, EDS1 may regulate JA simply due to the crosstalk between SA and JA signaling.

We have made the corresponding changes in the discussion.

4. The authors need to deposit their RNA-seq data to public domains for review and re-use of the data.

Raw sequencing data has been submitted to GEO (<http://www.ncbi.nlm.nih.gov/geo>) under accession number GSE115680.

Reviewer #2 (Remarks to the Author):

This paper proposes that *lux arrhythmo* (LUX) which is part of the evening complex of the Arabidopsis clock, plays a role in coordinating temporal defences in plants. The authors demonstrate this through pathogen assays with *Pseudomonas syringae* and *Botrytis cinerea*, salicylic acid measurements, RNA-seq analyses and ChIP assays. The data support the basic proposal, but there are some points that need clarification and/or correction. Although generally well-written, the manuscript suffers in a few places due to poor grammar and language use; this should be remedied. The findings are novel and do add to our understanding of temporal regulation of defence, as well as our understanding of clock function in plants. Interestingly, the authors propose that LUX can also act as an activator, but do not provide direct evidence for this. This seems to be an overinterpretation of the data. The authors do not comment on the role of other clock components that are affected in the loss-of-function *lux1* mutant. The paper might be enhanced by this analysis, perhaps in place of the EDS experiments which seem a little too peripheral to the main thrust of the paper?

We appreciate reviewer #2's comments on the novelty of this work. We have made major changes to this manuscript by including new data and more details of methods, rewriting of many parts of the manuscript, and making extensive editorial changes. Hope this reviewer is satisfied with these changes to improve the manuscript. Below we address this reviewer's other comments in this paragraph, point by point.

1) Interestingly, the authors propose that LUX can also act as an activator, but do not provide direct evidence for this. This seems to be an overinterpretation of the data.

We agree that our data are insufficient to conclusively demonstrate that LUX can activate gene expression. As per the reviewer's suggestion, we have taken caution in our interpretation of this role of LUX in the Discussion section.

" LUX likely acts as a transcriptional repressor to affect the expression of many target genes. It could also positively regulate gene expression (Figures 3 and 4B). Such a gene activation role of LUX could reflect LUX's repression of another repressor important for gene transcription. It is also possible that the LUX protein directly binds to some gene promoters, perhaps including that of *EDS1*, to enhance their expression."

2) The authors do not comment on the role of other clock components that are affected in the loss-of-function *lux1* mutant.

Expression of many clock genes are affected by *lux-1*. In particular, results from this report and other studies showed a direct binding of LUX to several core clock gene promoters, including *GI*,

LNK1, *LNK2*, *PRR7*, *PRR9*, and *LUX* itself^{4,5,6}. We discussed these genes and their functional relation to LUX in the Discussion section. We additionally tested a mutant allele of *ELF3*, encoding a clock protein interacting with LUX^{7,8}. Our results showed that both *elf3* and *lux* mutations conferred enhanced susceptibility to *Botrytis* infection. We postulate that individual clock genes or subdomains of the clock gene networks are linked to different output pathways and therefore demonstrate different phenotypes when the clock gene or the domain involving this gene is disrupted. Therefore, it would be interesting to further elucidate if some genes in the LUX subdomain, such as LUX clock targets and LUX interactors, could like LUX regulate similar output pathways, including plant defense. Details on this point please see the Discussion section.

3) The paper might be enhanced by this analysis, perhaps in place of the EDS experiments which seem a little too peripheral to the main thrust of the paper?

We think that the EDS1 story is important for this report. The reasons are as follows:

- i. *EDS1* is a well-known SA regulator;
- ii. we showed that *EDS1* is a direct target of LUX; and
- iii. we showed that *EDS1* is involved in JA signaling. Thus EDS1 helps to provide a mechanistic link of LUX in defense regulation. Interestingly although both loss of function in *EDS1* and *LUX* suppressed *acd6-1* phenotypes and SA accumulation, and enhanced disease susceptibility to *P. syringae*, the *eds1-2* mutant was not compromised in *Botrytis* resistance (Figure 2 and^{9,10}). Thus *LUX* acts partially through *EDS1* to affect SA signaling and/or JA signaling.

1. Figure 1. It would be helpful to show the levels of SA in wild-type plants (D) and the bacterial numbers in wild-type plants (F) too. Is the oscillation in SA as seen in the Goodspeed et al. (2012) study detectable in the wild-type (or *lux1*) – can't tell from G? What time (ZT) did the infection in G take place?

As suggested by this reviewer, we have showed SA levels and bacterial counts of Col-0 and *lux-1* in Figure 1D and 1F.

For basal SA levels, we only observed slightly higher values at 12 hpi in both Col-0 and *lux-1*. These higher SA levels are not significantly different from those at other time points without infection. SA accumulation is quite sensitive to the changes of environment based on our experience. Such a sensitivity is well supported by environment-dependent phenotypes displayed by some lesion mimic mutants, including *acd6-1* (Please refer to our response to a question raised by reviewer 3 regarding SA levels in the *acd6-1* plants). Thus we attribute the lack of detecting a clear oscillation of basal SA levels to our plant growth system, which may not be controlled with sufficient precision to allow us to detect the small differences in SA levels in a day.

The time (ZT1) used for infection is now indicated in Figure 1G legend.

2. Figure 2. What time did infection with *Botrytis cinerea* take place? What strain of *Botrytis*? How was the infection done – detached leaf? This information is not in methods.

We used *Botrytis cinerea* strain BO5-10, which was kindly provided by Tesfaye Mengiste at Purdue University. We used the whole plants for *Botrytis* infection. We have added details on *Botrytis* infection and disease scoring in the Methods section under Pathogen infection.

3. Figure 6. Please plot data in figures D and E on y-axes with same scales.

We modified the clock assay and presented the new data in Figures 5 and S7. Similar scales were used for related experiments.

4. Figure S2. What do uninfected *lux1* plants look like compared to wild-type? Do they start off with less chlorophyll? This is not convincing without this information.

We usually use 25-day old plants for infection experiments. At this stage and in the absence of pathogen challenge, Col-0 and *lux-1* look similar in terms of green color (Figure 1A). Upon infection of *P. syringae*, we observed increased chlorosis in *lux* mutants, compared with Col-0 and the *lux-4* rescue line (LUX-GFP) (Figures 2B right, 2D right, and S2). We clarified this point in the Results section.

5. Lines 89-104 are inappropriate for the Introduction. The paragraph in lines 109-118 is more suitable.

We have modified these two paragraphs according to this comment.

6. Lines 75-76: this is not true. Goodspeed et al. (2012) did not demonstrate that *lux* and *cca1* affected susceptibility to insects. They used the *lux2* mutant and the CCA1-ox line as arrhythmic plants to demonstrate that clock function was responsible for the phase dependent resistance. This should be removed.

We have modified this statement in the Introduction according to this comment.

7. Line 178: remove 'a' from 'time of a day'

We removed 'a' from 'time of a day'.

8. Lines 194-198: References should be provided for the expected figures of cycling genes. Furthermore, the way that this is worded makes it difficult to understand what the authors mean. Do they mean that of the genes the current study found to be affected by LUX, 26.7% had previously been demonstrated to cycle under LD conditions, and a further 26.3% to cycle under LL conditions? Or are they saying that of the genes the current study found to be affected by LUX, only 26.7% and 26.3% respectively had previously been demonstrated to cycle under LD or LL?

We rewrote this section as the follow and hopefully we have now made our points clearer.

" To determine *LUX*-affected genes that also oscillate in a day, the web-based tool Phaser^{11, 12}

was used to analyze gene expression in LD and LL, using publicly available microarray data^{13,14}. Of the 1618 *LUX*-affected transcripts, 26.7% genes cycled under LD and 26.3% cycled under LL. When we analyzed the entire Arabidopsis genome, we found that 18.9% of the transcripts cycled in LD and 17.8% in LL. This observation of enrichment of cycling transcripts in the set of *LUX*-affected transcripts suggests that *LUX* preferentially regulates expression of cycling genes, consistent with *LUX* being a core clock regulator. Because only two time points (ZT1 and ZT13) were used in our RNAseq analysis, we may have missed some cycling genes that are affected by *LUX* at other times of day."

9. Lines 219-221: The expression of group III genes generally being lower in *lux1* does not necessarily equate to *LUX* having transcription activating activity. The authors need to consider that lack of *LUX* may lead to the reduction of another repressor, which then results in upregulation of some genes. This assertion is repeated in line 380 and should be treated with caution.

We agree with this reviewer that we should take caution when asserting *LUX* as a transcriptional activator. Please see our answer above to this question (at the beginning of this reviewer's comments).

10. Lines 265-267: Goodspeed et al. (2012) did not demonstrate that *lux* affected susceptibility to cabbage loopers. They used the *lux2* mutant as an arrhythmic line (as well as the *CCA1-ox* arrhythmic line) to demonstrate that clock function was responsible for the phase dependent resistance. This should be corrected in line 373 too.

We have modified relevant statements in the manuscript according to this comment.

11. Rephrase to make clearer: lines 509-510 "To determine if a *LUX*-affected gene cycles during a day, the web-based tool Phaser was used to analyze gene expression under one diurnal..."

We have revised this sentence. Please see our answer to Q8 raised by this reviewer.

Reviewer #3 (Remarks to the Author):

In this manuscript, Zhang et al. describe a role for the Arabidopsis clock transcription factor *LUX ARRHYTHMO* on the regulation of plant defense responses. Authors indicate that *lux-1* mutant plants have compromised disease resistance against *P. syringae* and botrytis infections and both SA and JA signaling. Using RNAseq *LUX* downstream target genes are identified. In particular, *LUX* was confirmed to bind to the promoters of *EDS1* and *JAZ5*, which are involved in the SA and JA signaling pathways respectively. Finally, authors found that coronatine (a bacterial chemical that mimics some JA functions) but not MeJA treatments resulted in a longer clock period phenotype.

In the present format the manuscript is too preliminary and needs to be refocused to answer a specific question. If authors are attempting to establish a role for *LUX* as a coordinator of clock and defense responses, then experiments should be performed in constant conditions. However, it seems that LD was the condition used for most experiments, thus results reflect the role of

LUX on defense regulation either by light/dark changes and the clock, or just light/dark changes (if this is the case then the interpretation of most results should be reconsidered). For example, while infections in LD (figure 2A) indicate that *lux-1* plants are more sensitive to *P. syringae*, infections in LL do not show any difference suggesting that LUX specific function depends on the presence of LD cycles. In fact, authors observed that, differential susceptibility to ZT1 versus ZT13 infections in LL is identical in wild type and *lux-1* plants. Regarding this later experiment LL “morning” infections should have been performed at ZT25 rather than ZT1 (as only after ZT12 plants are in free running conditions). Thus, overall experiments presented here indicate that LUX mediates defenses (*P. syringae* and *botrytis*) in light/dark cycles rather than the clock.

Please see our answer to a related question raised by reviewer 1 (major question #1).

Regarding the mechanisms, it seems that LUX regulates callose deposition upon flagellin perception, however results are far less convincing regarding its function on SA production in the absence of the *acd6-1* mutation (Fig. 1G).

We respectfully disagree with this reviewer's interpretation of the results. Certainly, the reduction of callose deposition in *lux-1* in response to *flg22* and *HrcC* is dramatic (Fig 2F), supporting the role of LUX in regulating basal defense. For LUX's role in SA regulation, we believe that we have also strong supporting evidence as summarized below:

- 1) *lux-1* suppresses high SA accumulation besides other phenotypes in *acd6-1* (Figure 1A-1F).
- 2) *lux-1* is compromised in acute SA accumulation in *P. syringae* infected tissue (Figure 1G).
- 3) *lux* mutations allow increased bacterial growth (Fig 2B) and impaired SAR (Fig 2D), both phenotypes are related to SA production and signaling.
- 4) RNAseq analysis reveals that many SA-related genes are affected by *lux-1* (Figure 3A).
- 5) the main SA regulator, *EDS1*, is a direct transcriptional targets of LUX (Figure 3B).

We believe these data convincingly show a role of LUX in SA regulation, at least partially acting through *EDS1*. We summarized these points in the Discussion section and hopefully we have now better clarified the role of LUX in SA regulation.

RNAseq experiments should have been extended to all LUX regulated genes, not specific “lists” of genes. Furthermore, the identification of novel LUX “TTFL” (see comment below) target genes, although interesting, is not relevant for this manuscript. More importantly, follow up experiments on identified genes, such as *EDS1* and *JAZ5*, should be included to establish their role in mediating LUX regulation of defense responses.

This is a good suggestion; accordingly, we performed heatmap analysis of all 1618 LUX-affected genes. We deleted the analysis on lists of clock and defense genes. The new analysis is presented as Figure 3A. Thanks!

We again respectfully disagree that the identification of novel LUX “TTFL” is not relevant for this manuscript. The identification of additional clock TTFL targets (*LNK2* and second LUX-binding site on *LNK1* promoter) and the output gene *CDF1* provides new information on the role

of LUX in clock regulation. These data expand the clock gene networks that require a direct functional input of LUX and also show that LUX can execute a direct control of the output pathways. Thus our study further illustrates the complexity of clock TTFL regulation.

We have better clarified the role of EDS1 in LUX-mediated in the related Result and Discussion sections. Briefly, *EDS1* is a major SA regulator involved in the SA-signal amplification loop^{9,15}. Loss of function in both *EDS1* and *LUX* leads to similar phenotypes in suppression of *acd6-1*, reduced SA accumulation upon pathogen challenge, and *P. syringae* susceptibility. Thus *LUX* could act through *EDS1* in regulated SA-mediated defense. The suppressed expression of some other SA genes and SA accumulation in *lux-1* could be due to reduced *EDS1* expression in *lux-1*. We further provide data on the role of EDS1 in other LUX-mediated defense phenotypes, including *Botrytis* response, JA sensitivity, and JA signaling under pathogen challenge (Figure 2C, 4A, and 4C). These data reveal a JA-regulatory function of EDS1, which was previously unknown. Interestingly, the *eds1-2* mutant was not compromised in *Botrytis* resistance (Figure 2C). Therefore, we conclude that *LUX* only acts partially through *EDS1* to affect SA signaling and/or JA signaling.

JAZ5 is a gene from a large gene family and the single mutants of the family members mostly do not show defense defects. Therefore, it would be difficult to use a loss of function approach, as we did with *EDS1*, to study the role of *JAZ5* in *LUX*-mediated defense. However, we do intend follow up with *JAZ5*, using other approaches, such as gain of function approaches and multiple *JAZ* knockouts, to assess *JAZ5* and its homologs in *LUX*-mediated defense in the future.

Finally authors explored the regulation of Arabidopsis clock function by MeJA and coronatine. I find that these experiments are very preliminary and not conclusive regarding the role of MeJA in clock regulation (and probably for coronatine as well). To properly establish if MeJA does or does not have a role in clock regulation, a series of experiments should have been performed treating plants at different times of the day (this is critical as most clock responses are gated at specific times of the day). If MeJA indeed does not regulate the clock function, what about JA-Ile?. More importantly, given that authors propose that *LUX* mediates JA responses, does *LUX* mediate clock responses to coronatine?.

Please see our answer to the related question raised by reviewer 1 (major question #3).

Specific comments:

All text sections require editing.

We have made extensive changes to the entire manuscript and hope that this reviewer finds the manuscript to be much improved.

Additional experimental details should be added both to the materials and methods sections and legends for main figures. (I found the legends for supplementary figures much more informative than those for main figures)

We have added more details in the Methods section and figure legends.

The use of the term “TTFL genes” to refer to clock genes is rather confusing as TTFLs are not exclusive for the clock function.

We have changed TTFL genes to either clock TTFL genes or clock genes through the text.

How were plant sizes determined in Fig. 1B?

As we did in the past for similar plant size quantitation, we measured plant size for the largest distance between tips of two rosette leaves, using at least 20 plants for each genotype. This is now indicated in Figure 1 legend.

Total SA levels reported in Fig 1D for the *acd6-1* mutant are ~10 times higher than what authors published before in Zhang et al. 2013. What would be the reason for such difference?

We thank this reviewer to bring our attention to the difference in the SA levels between this report and one publication in our lab. We went back to check the SA levels in several other publications from our laboratory. Of seven publications from 2009 to 2016, we found that four papers showed 65 -100 $\mu\text{g/g}$ FW total SA in *acd6-1*^{9,16,17,18}, two showed 30-40 $\mu\text{g/g}$ FW total SA^{19,20}, and the lowest one 12 $\mu\text{g/g}$ FW total SA in Zhang et al 2013 paper²¹. The SA levels reported in this manuscript are around 90 $\mu\text{g/g}$ FW total SA, reasonably consistent with most previously published values.

We went back to check the original data for the Zhang et al 2013 paper and did not find any problems with data calculation. We have no good explanation for why those values were low relative to others and would like to attribute the low SA levels in Zhang et al 2013 paper to subtle changes of growth conditions, however unsatisfying that explanation might be. However, we note that many lesion mimic mutants have been shown environment-dependent phenotypes. Like other lesion mimic mutants, *acd6-1* is sensitive to changes of light, humidity, and temperature. Slight changes in these conditions cause variations in *acd6-1* phenotypes (including changes in SA levels, plant size, and cell death severity). Humidity is especially tricky—even uncovering seedlings at different times in different experiments can affect *acd6-1* phenotypes.

RNA blots (fig. 1E) should be quantified (or better, PR1 levels determined by Q-PCR). If LUX down regulates SA signaling then PR1 expression should be tested in *col-0* and *lux-1* plants upon SA treatment.

We replaced Figure 1E with qRT-PCR data, as suggested.

Bacterial titers should be normalized to 1 mm²

We normalized bacterial titers to 1 cm², which appears to be more commonly used than 1 mm².

Authors state the SA levels oscillate during the day (line 110), however this does not seem to be the case according to the results in figure 1G.

In Figure 1G, we saw slightly increase SA levels at 12 hpi in both mock-treated *Col-0* and *lux-1*,

although these values are not significantly different from the values at other time points for non-pathogen treated samples. Our failure to detect an oscillation in basal SA levels could reflect low basal expression associated with environmental conditions, as we discussed above regarding SA levels in *acd6-1*.

Images in Fig. 2C don't seem representative of the results shown in Fig. 2D (at least for *lux-1*). To better support these results the comparison between treated and untreated plants for each genotype should be presented.

The image in Fig. 2C (now Figure 2E right) is from one experiment and 2D (now Figure 2E left) is the average ratio of three independent experiments. We have clarified this in the figure legend. We think it is better to quantify the average ratio from three independent experiments than to only show the treated and untreated plants for each genotype from one experiment. Thus, it may be difficult to find an image to exactly reflect the average ratio. Nevertheless, we changed the image to a new one, which should be slightly better at reflecting the average ratio.

Callose deposition quantification should be normalized per mm² (Fig. 2F)

We have normalized callose deposition per mm².

Text references to the manuscript by Goospeed et al (2012) should be revised, as this manuscript indicated that the overall clock function (rather than specifically LUX) regulates plant herbivory resistance.

We have modified relevant statements in the manuscript according to this comment.

ZT values in figures 6D and 6E are incorrect. ZT or “zeitgeber time” provides a reference to the last dark to light transition.

We have corrected this in the new figure 5 and figure S7, and have altered the axis to read simply Time in LL (h).

Regarding luciferase assays:

What was the light intensity in LL?

The luciferase assay light intensity is 10 $\mu\text{mol m}^{-2} \text{s}^{-1}$ photon flux density in LL. We also indicated carefully light intensity for plant growth and other assays in the related Result and Method sections.

Plants were grown in LD and then treated with cor or MeJA. Either to perform the treatments or after the treatment these plants had to be moved to a different plate. How did root damage was prevented in this process? I wonder if plant manipulation and/or tissue damage (which could result in JA production) had an effect on clock rhythms. A set of plants that are not treated or manipulated should be processed in parallel to address this potential issue.

It is indicated that seedlings were “briefly” soaked in coronatine or MeJA solutions (the length of treatment should be provided).

The time of day at which treatments were performed should be indicated.

We were very careful when transferring plants to 96-well plates to minimize root damage. Transferred seedlings stayed in 1 d in LD and 1d in LL before being treated with chemicals and measured for clock activity. Mock-treated plants were included on the same plate as the control. For treatments, we now add a chemical solution directly to the seedlings in a 96-well plate to minimize variation in exposure time for individual seedlings. Time of day for the treatments are now indicated in the figures, figure legends, and text (Results and Methods sections). For additional information regarding the clock assays, please see our answer to major question #3 raised by Reviewer 1.

Fig S2A, axis label is missing.

We added the label to Figure S2A.

Reviewer #4 (Remarks to the Author):

The authors demonstrated comprehensive studies to understand molecular mechanisms underlying relationship between circadian clock and innate immunity in Arabidopsis. They found that *lux1-1* mutants suppressed constitutive defense phenotypes of *acd6-1*. Phenotypic and genome-wide gene expression analyses of *lux-1-1* suggested possible molecular mechanism of LUX for innate immunity. Finally, they showed that strong antagonist of JA, coronatine, can modulate circadian period.

The paper provides potentially interesting results for understanding of molecular link between clock and innate immunity. However, I have concerns about experimental design, data, and interpretation of results.

The authors used only *lux-1* allele in this study. While we know that *lux-1* and other *lux* alleles show same circadian phenotypes (e.g., Hazen et al., PNAS 2005), it is still unknown whether immunity-related phenotypes of *lux-1* are allele specific (Figure 1, 2). Immunity-related phenotypes of other *lux* alleles or genetic complementation assay are required to conclude that LUX regulates immunity. I also wonder if *elf3* and *elf4* show immunity-related phenotypes, because LUX, ELF3, and ELF4 form Evening Complex. RNAseq data of *lux-4*, *elf3* and *elf4* mutants might be useful.

We appreciate this reviewer's recognition of the importance of our work in understanding of molecular link between clock and innate immunity. As per reviewer's suggestion, we have now provided pathogen response data in LL, and expanded our study to include mutants carrying two distinct *lux* alleles, a *lux-4* complementation line (LUX-GFP), Col-0, and other genotypes (Figures 2B and 2C). These data support that *LUX* is a positive regulator of plant resistance against the bacterial pathogen *P. syringae* and the fungal pathogen *Botrytis*.

An *elf3* allele was previously shown to be compromised to resistance to *P. syringae*²² and *B. cinerea* with detached leaf assay²³. We presented new data in this revision that the *elf-7* allele

was more susceptible to *Botrytis* infection, using a whole plant assay.

This study is to elucidate the role of LUX in regulating clock and defense. Thus the RNAseq analysis is focused on *lux-1*. We fully agree with the suggestion that future experiments should include RNAseq data of *lux-4*, *elf3* and *elf4* mutants, but we feel that they lie beyond the scope of the present study. We appreciate the suggestion!

The authors claimed that LUX regulates immune responses through EDS1. I felt this is very likely. However, genetic study using *lux1/EDS1* overexpression is useful to consider whether EDS1 has major role in LUX-dependent immune regulation.

The reviewer raised a great question regarding whether EDS1 has a major role in LUX-dependent immune regulation. We believe that we have clarified this role of EDS1 in LUX-mediated defense in the related Result and Discussion sections. Briefly, *EDS1* is a major SA regulator involved in the SA-signal amplification loop^{9, 15}. Loss of function in both *EDS1* and *LUX* leads to similar phenotypes in suppression of *acd6-1*, reduced SA accumulation upon pathogen challenge, and *P. syringae* susceptibility. Thus *LUX* could act through *EDS1* in regulated SA-mediated defense. The suppressed expression of some other SA genes and SA accumulation in *lux-1* could be due to reduced *EDS1* expression in *lux-1*. We further provide data on the role of EDS1 in other LUX-mediated defense phenotypes, including *Botrytis* response, JA sensitivity, and JA signaling under pathogen challenge (Figures 2C, 4A, and 4C). These data revealed the JA-regulatory function of EDS1, which was previously unknown. Interestingly, the *eds1-2* mutant was not compromised in *Botrytis* resistance (Figure 2C). Therefore, we conclude that *LUX* only acts partially through *EDS1* to affect SA signaling and/or JA signaling.

The use of *lux1/EDS1* overexpression may help to confirm some aspects of *EDS1*'s role in *LUX*-mediated defense as described above. The use of a gain of function version of *EDS1* in the *lux* background could also complicate the interpretation of some results. An alternative approach would be to make the *lux-1 eds1-2* double mutant. We have begun to generate these additional *LUX* and *EDS1* related genetic materials. But given the time required to generate these mutants, we think that characterization of these plant materials is beyond the scope of this report.

Expression changes of LUX-target genes previously reported (e.g., PRR9, PRR7, and GI, from Ezer et al., Nature Plants 2017) in *lux-1* seem too little (Figure 3). This figure is not convincing to support that LUX is involved in clock and immune response. I recommend other data presentation styles to say LUX control clock genes, but this was already done by previous studies (Helfer et al., Curr. Biol., 2011, Ezer et al., Nature Plants 2017).

We agree with this reviewer that a number of prior studies have established that LUX controls clock gene expression as well as other pathways. In this report, our focus was to extend this to explore the potential of LUX to regulate defense genes (new Figure 3A and Table 1). The bioinformatics analysis of LUX-affected gene promoters for the LBS motif followed by ChIP experiments revealed new transcriptional targets of LUX related to the circadian clock, including LNK2, the second LUX-binding site on LNK1 promoter, and the output gene CDF1. These data expand the clock gene networks that require a direct functional input of LUX and also show that

LUX can execute a direct control of the output pathways.

For the discrepancy in the quantity of some gene affected by LUX between our report and some studies reported previously, we would like to point out that we only used two time points in the RNAseq experiment, which may miss the peak expression of many genes. We confirmed our RNAseq data with qRT-PCR with some selected clock and defense genes. Therefore, we believe that our data support LUX function in clock and defense regulation.

I think the data represented in Figure 6 are too preliminarily and necessary for this manuscript. First, effects of COR was not so strong. Indeed, clock period was lengthened with very high concentration of COR (100 μ M) than that used in immunity papers (low μ M range), suggesting that COR effect on clock is artifact. Light and temperature, two major signals regulating the clock, can alter period length more significantly. How many potent JA analogues were tested, and how many compounds affect the period? Do these compounds really affect only JA signaling in plants? Did knock-down of JA signaling genes alter circadian period? In addition, COR effects on clock TTFL genes were very small (Figure S 8). Further experiments are needed to conclude that COR regulates clock.

We agree with the reviewer's comments and appreciate the suggestions for improving our experiments. According to this reviewer and other reviewers' suggestions, we conducted more extensive experiments to investigate the role of JA in clock regulation. Our data support a reciprocal regulation of the circadian clock by JA signaling activation. Details related to this topic please see our answer to major question #3 raised by reviewer 1.

minor comments.

We do not use ZT for time value for samples analyzed under constant light conditions. Instead, time in constant light (h) is used.

We have changed Figure 5 and Figure S7 in response to the reviewer's suggestion. Thank you!

LUX binds to LNK1 promoter (Mizuno et al., *Plant Signal Behav.*, 9, e28505).

We have included this reference in the revised manuscript. Thanks for pointing this out! Mizuno et al showed that LUX binds to the region slightly after the transcription starting site⁶, which is around LNK1-3' in Figure 3D. Our bioinformatics analysis did not detect this LBS site (LNK1-3') because we only analyzed sequence within the 1500 bp-promoter region, relative to the transcription start site of each gene. It is worth noting that without prior knowledge of this binding site, we reported LUX binding to both LNK1-3' and LNK1 sites. Thus our ChIP experiment was validated.

Definition of clock-related genes is poor, though the authors mentioned that these genes were from a list of genes involved in rhythmic processes in TAIR website. Does clock-related genes contain clock-output genes? If so, they contain lots of genes not involved in core clock function.

The clock-related genes include both core clock genes and output genes in this report. Per

reviewers #3 and #4's suggestions (see above related questions), we no longer focus on the expression of a list of clock genes. Instead, we used all LUX-affected genes (1618) for the heatmap analysis (new Figure 3A). Therefore, we deleted the supplementary table containing these genes and revised relevant figures.

References:

1. Hickman R, *et al.* Architecture and Dynamics of the Jasmonic Acid Gene Regulatory Network. *Plant Cell*, (2017).
2. Attaran E, *et al.* Temporal Dynamics of Growth and Photosynthesis Suppression in Response to Jasmonate Signaling. *Plant Physiol* **165**, 1302-1314 (2014).
3. Anders S, Huber W. Differential expression analysis for sequence count data. *Genome Biol* **11**, R106 (2010).
4. Helfer A, Nusinow DA, Chow BY, Gehrke AR, Bulyk ML, Kay SA. LUX ARRHYTHMO encodes a nighttime repressor of circadian gene expression in the Arabidopsis core clock. *Curr Biol* **21**, 126-133 (2011).
5. Mizuno T, *et al.* Ambient temperature signal feeds into the circadian clock transcriptional circuitry through the EC night-time repressor in Arabidopsis thaliana. *Plant Cell Physiol* **55**, 958-976 (2014).
6. Mizuno T, Takeuchi A, Nomoto Y, Nakamichi N, Yamashino T. The LNK1 night light-inducible and clock-regulated gene is induced also in response to warm-night through the circadian clock nighttime repressor in Arabidopsis thaliana. *Plant Signal Behav* **9**, e28505 (2014).
7. Herrero E, *et al.* EARLY FLOWERING4 recruitment of EARLY FLOWERING3 in the nucleus sustains the Arabidopsis circadian clock. *Plant Cell* **24**, 428-443 (2012).
8. Dai S, *et al.* BROTHER OF LUX ARRHYTHMO is a component of the Arabidopsis circadian clock. *Plant Cell* **23**, 961-972 (2011).
9. Ng G, Seabolt S, Zhang C, Salimian S, Watkins TA, Lu H. Genetic dissection of salicylic acid-mediated defense signaling networks in Arabidopsis. *Genetics* **189**, 851-859 (2011).
10. Falk A, Feys BJ, Frost LN, Jones JD, Daniels MJ, Parker JE. EDS1, an essential component of R gene-mediated disease resistance in Arabidopsis has homology to eukaryotic lipases. *Proc Natl Acad Sci U S A* **96**, 3292-3297. (1999).
11. Mockler TC, *et al.* The DIURNAL project: DIURNAL and circadian expression profiling, model-based pattern matching, and promoter analysis. *Cold Spring Harb Symp Quant Biol* **72**, 353-363 (2007).
12. Michael TP, *et al.* Network discovery pipeline elucidates conserved time-of-day-specific cis-regulatory modules. *PLoS Genet* **4**, e14 (2008).

13. Smith SM, *et al.* Diurnal changes in the transcriptome encoding enzymes of starch metabolism provide evidence for both transcriptional and posttranscriptional regulation of starch metabolism in *Arabidopsis* leaves. *Plant Physiol* **136**, 2687-2699 (2004).
14. Harmer SL, *et al.* Orchestrated transcription of key pathways in *Arabidopsis* by the circadian clock. *Science* **290**, 2110-2113 (2000).
15. Bartsch M, *et al.* Salicylic acid-independent ENHANCED DISEASE SUSCEPTIBILITY1 signaling in *Arabidopsis* immunity and cell death is regulated by the monooxygenase FMO1 and the Nudix hydrolase NUDT7. *Plant Cell* **18**, 1038-1051 (2006).
16. Hamdoun S, *et al.* Differential Roles of Two Homologous Cyclin-Dependent Kinase Inhibitor Genes in Regulating Cell Cycle and Innate Immunity in *Arabidopsis*. *Plant Physiol* **170**, 515-527 (2016).
17. Wang G, Zhang C, Battle SL, Lu H. The Phosphate Transporter PHT4;1 is a Salicylic Acid Regulator Likely Controlled By the Circadian Clock Protein CCA1. *Frontiers in Plant Science* **5**, (2014).
18. Lu H, *et al.* Genetic analysis of *acd6-1* reveals complex defense networks and leads to identification of novel defense genes in *Arabidopsis*. *Plant J* **58**, 401-412 (2009).
19. Wang GF, Seabolt S, Hamdoun S, Ng G, Park J, Lu H. Multiple roles of WIN3 in regulating disease resistance, cell death, and flowering time in *Arabidopsis*. *Plant Physiol* **156**, 1508-1519 (2011).
20. Wang GY, Shi JL, Ng G, Battle SL, Zhang C, Lu H. Circadian clock-regulated phosphate transporter PHT4;1 plays an important role in *Arabidopsis* Defense. *Mol Plant* **4**, 516-526 (2011).
21. Zhang C, *et al.* Crosstalk between the circadian clock and innate immunity in *Arabidopsis*. *PLoS Pathog* **9**, e1003370 (2013).
22. Bhardwaj V, Meier S, Petersen LN, Ingle RA, Roden LC. Defence responses of *Arabidopsis thaliana* to infection by *Pseudomonas syringae* are regulated by the circadian clock. *PLoS One* **6**, e26968 (2011).
23. Ingle RA, *et al.* Jasmonate signalling drives time-of-day differences in susceptibility of *Arabidopsis* to the fungal pathogen *Botrytis cinerea*. *Plant J* **84**, 937-948 (2015).

Reviewers' comments:

Reviewer #1 (Remarks to the Author):

The revised manuscript by Zhang et al. has addressed most of my previous concerns. There are still a few points left:

1. Lines 94-96: As the authors also mentioned in the discussion part, Psm avrRpt2 induced systemic SA level increase has been shown to be dependent on CHE, a clock gene. Therefore it is not appropriate here to say "no clock genes have yet been reported so far to play such a role in SA regulation"
2. Fig. S3B: missing connectors linking Col-0.1, Col-0.2, Col-0.3
3. Line 234: base on the GO analysis, it appears that the clustering analysis did not generate much differences in the three groups.
4. Line 236: "and II" should be "and III"?
5. Line 240: group II is also enriched in genes responding to abiotic and biotic stimuli
6. Lines 249-250: the qPCR results seem do not support statistically significant changes of MYC2
7. Lines 272-276: it is quite strange that the authors used Pma instead of Botrytis to study the response of JA marker genes since Pma is usually used as a model of biotrophic pathogen while JA is more involved in defence against necrotrophic pathogens.
8. Reference number 1 and 17 are the same paper.

Reviewer #2 (Remarks to the Author):

You have addressed my concerns, and I am satisfied with the revisions

Reviewer #3 (Remarks to the Author):

In this revised version, authors included additional experiments and reformatted/edited many sections of the manuscript. While I appreciate authors' efforts to improve the manuscript I still think that this work does not establish a role for LUX as a "coordinator of clock and defense responses". As I mentioned before, most experiments were performed in LD, suggesting a role for LUX in regulating defenses in light/dark cycles. In fact LUX (and the EC) was shown to regulate light signaling. New infection experiments in LL seem to indicate that this is the case, as defense phenotypes were only observed when plants were grown under a very low light intensity (typically not used in clock experiments). Furthermore, experiments in figure 2B indicate that lux mutant plants are more resistant to infections at ZT13 compared to ZT1, indicating that clock regulation of defenses against *P. syri* was not affected in the lux mutant background. Increased susceptibility to the pathogen was observed after both ZT1 and ZT13 infections suggesting that the overall defense was affected.

Looking at differentially expressed genes a number of LUX regulated genes was uncovered. Given that in these experiments sampling was done at only two time points (ZT1 and ZT13) and that most LUX targets exhibit daily rhythms, it is imperative to address if the phase of expression of rhythmic genes is the same in all genetic backgrounds (wt, lux, acd6 and lux/acd6) used in the experiment (different phases in each background would mislead the identification of differentially expressed genes). In addition, as I mentioned before follow up experiments (i.e. using genetics) are needed to support the role of identified genes, such as EDS1 and JAZ5, in mediating LUX regulation of defense responses.

Finally, I find that experiments that attempted to establish clock regulation by JA, MJ or JA-Ile should be further revised. First, I find intriguing that the period length under LL (10uE) was close to 24h. As previously shown, under such low light intensity the period should have been significantly longer. Second, period length and phase phenotypes were observed with only the GRP7:LUC reporter. While authors indicate that this reporter may be more sensitive to the

hormone treatment, this result indicates that either two clocks are functioning simultaneously with a different period or that two different tissue specific clocks can run with a different period. Such result would be highly novel but would require further supporting experiments. Third, the amplitude of CCA1 and GRP7 rhythms are consistently reduced in a dose dependent manner. However, JA was reported to significantly reduce plant growth in a dose dependent manner, which could have biased the amplitude phenotype when using a luciferase reporter. Normalization by plant size, or other methods (i.e. gene expression by qPCR) could be considered. Finally, if LUX is a coordinator of clock and defense responses, what would be the role of LUX in mediating JA, MJ or JA-Ile regulation of the clock function?

Reviewer #4 (Remarks to the Author):

The authors demonstrated comprehensive studies to understand molecular mechanisms underlying relationship between circadian clock and innate immunity in Arabidopsis. They found that lux mutants suppressed constitutive defense phenotypes of acd6. Phenotypic and genome-wide gene expression analyses suggested possible molecular mechanism of LUX for innate immunity through ACD6. They showed that strong antagonist of JA signaling can modulate circadian clock.

It was clear that LUX is involved in defense response, since the authors further analyzed two independent lux alleles and elf3 to confirm that LUX (and ELF3, an interaction of LUX) is involved in defense response. ChIP-qPCR experiment with appropriate control loci confirmed that LUX associates with JAZ5 and EDS promoters.

However, I have still concern about the conclusion that JA signaling controls the clock, which seems to be proposed by Fig. 5 and Supplemental Fig. 6 and 7. I appreciated the authors effort to consider if JA signaling controls the clock. However, the data presented here were not convincing to support their propose. The authors found that amplitude of both morning and evening reporters (CCA1:LUC and GPR7:LUC) were decreased, but period length were not changed. This suggests that overall plant vigorousness or activity was decreased upon JA treatment, but does not suggest circadian clock is controlled by JA signaling. Again, even though such decreased amplitude, I see that most crucial parameter of the clock, period length, were not drastically changed, suggesting circadian clock is robust against to JA treatment.

Responses to Reviewers' Comments:

Reviewer #1 (Remarks to the Author):

The revised manuscript by Zhang et al. has addressed most of my previous concerns. There are still a few points left:

1. Lines 94-96: As the authors also mentioned in the discussion part, Psm avrRpt2 induced systemic SA level increase has been shown to be dependent on CHE, a clock gene. Therefore it is not appropriate here to say “no clock genes have yet been reported so far to play such a role in SA regulation”

Here we meant to say that no clock genes have yet been reported so far to play such a role in regulating "acute SA accumulation in the local infected region". We clarified this point in the revised text. CHE was only shown to affect SA levels in the SAR tissue, which are much lower than those induced in the local infected tissue. CHE has not been demonstrated to affect local SA accumulation with *P. syringae* infection.

2. Fig. S3B: missing connectors linking Col-0.1, Col-0.2, Col-0.3

Connectors linking Col-0.1, Col-0.2, Col-0.3 are shown now.

3. Line 234: base on the GO analysis, it appears that the clustering analysis did not generate much differences in the three groups.

There are some differences among the three groups as we described in the text (also see below).

"Cluster analysis of the 1618 *LUX*-affected genes revealed three major groups (Figure 3A). Expression of many genes in group II was relatively low in all four genotypes, compared with those in groups I and III. Some genes in group II showed greater expression in *lux-1*, supporting a known role of *LUX* as a transcriptional repressor. Most genes in group I were highly induced in *acd6-1*. Expression of most group I and III genes was suppressed by *lux-1*, especially at ZT1 and/or in the *acd6-1* background, suggesting that *LUX* can also positively affect gene expression via direct or indirect means. GO analysis revealed that groups I and III were more enriched than group II in genes responding to abiotic and biotic stimuli (Figure 3A)."

4. Line 236: “and II” should be “and III”?

This is corrected now.

5. Line 240: group II is also enriched in genes responding to abiotic and biotic stimuli

We modified the text to reflect this point as the follow:

"GO analysis revealed that groups I and III were more enriched than group II in genes responding to abiotic and biotic stimuli."

6. Lines 249-250: the qPCR results seem do not support statistically significant changes of MYC2

There is no statistical significance in MYC2 expression among the four genotypes. The letters "b" were mislabeled. We made the correction. We appreciate the reviewer noticing our error.

7. Lines 272-276: it is quite strange that the authors used Pma instead of Botrytis to study the response of JA marker genes since Pma is usually used as a model of biotrophic pathogen while JA is more involved in defence against necrotrophic pathogens.

The point of this paragraph and the related Figure 4B and 4C is to show that LUX and EDS1 could affect JA signaling under defense conditions. Both Pma and Botrytis can activate host defense and are known to affect JA signaling during infection. We did observe the effect of LUX and EDS1 on JA gene expression with Pma infection (Figure 4B and 4C).

The reason that we prefer using Pma rather than Botrytis for a pathogen-induced gene expression study is that Pma-infected plants are kept in the same growth condition used for plant growth. They show nice disease symptoms and gene expression responses. There is no need to cover the plants. In the case of Botrytis infection, the fungus needs high humidity to infect plants well and, therefore, Botrytis-infected plants are covered during the infection process. We feel that high humidity may complicate gene expression results in some genetic backgrounds. Nevertheless, we agree with this reviewer that gene expression analysis with Botrytis infection has been done and is an alternative way to gauge the changes in expression of JA genes and other defense genes.

8. Reference number 1 and 17 are the same paper.

We corrected this mistake.

Reviewer #2 (Remarks to the Author):

You have addressed my concerns, and I am satisfied with the revisions

Reviewer #3 (Remarks to the Author):

In this revised version, authors included additional experiments and reformatted/edited many sections of the manuscript. While I appreciate authors' efforts to improve the manuscript I still think that this work does not establish a role for LUX as a "coordinator of clock and defense responses". As I mentioned before, most experiments were performed in LD, suggesting a role for LUX in regulating defenses in light/dark cycles. In fact LUX (and the EC) was shown to regulate light signaling. New infection experiments in LL seem to indicate that this is the case, as defense phenotypes were only observed when plants were grown under a very low light intensity (typically not used in clock experiments). Furthermore, experiments in figure 2B indicate that lux mutant plants are more resistant to infections at ZT13 compared to ZT1, indicating that clock regulation of defenses against *P. syri* was not affected in the lux mutant background. Increased

susceptibility to the pathogen was observed after both ZT1 and ZT13 infections suggesting that the overall defense was affected.

We respectfully disagree with this reviewer's interpretation of our data for the role of LUX in defense regulation.

1) We show in this report that *LUX* loss of function plants are more susceptible to *P. syringae* and *Botrytis* in a free running condition (LL with $10 \mu\text{mol m}^{-2} \text{s}^{-1}$ photon flux density) (Figure 2B and 2C) and that resistance is rescued by the functional *LUX-GFP* gene. Thus we provide critical data to demonstrate the role of *LUX*-mediated clock in defense regulation. Plants are known to employ different mechanisms to fight against pathogens and pests with different lifestyles at different times of day. For epiphytic bacterial pathogens, such as spray-infected *P. syringae*, stomata-independent defense is strong during daytime while stomata-dependent defense is dominant at night¹. The *lux* mutants are more resistant to spray-infection of *P. syringae* in the subjective evening, compared to that performed at subjective morning (Figure 2B). These results suggest that *LUX*-mediated circadian clock only partially affects stomata-dependent defense in the evening in response to epiphytic *P. syringae* and additional factors also contribute to plant immunity at night. On the other hand, for the necrotrophic fungal pathogen *Botrytis*, plants use different defense mechanisms and are less dependent on stomatal-defense. Accordingly, *lux* mutants infected with *Botrytis* in the subjective evening are not more resistant than those infected in the subjective morning (Figure 2C).

2) While establishing that *LUX*-mediated circadian clock regulates plant innate immunity, enhanced disease susceptibility of the *lux* mutants is shown in LL with light intensity of $10 \mu\text{mol m}^{-2} \text{s}^{-1}$ photon flux but not of $180 \mu\text{mol m}^{-2} \text{s}^{-1}$ photon flux. These results suggest that the defense role of the circadian clock is conditional and influenced by light. The EC, consisting of *LUX-ELF3-ELF4*, is known to regulate light signaling through affecting expression of many photosynthesis genes and light response genes². Thus, mutations in *LUX* or other EC genes could make plants particularly sensitive to light. Consistent with this, *LUX* was shown to be a repressor of leaf senescence³. Although not supporting higher *P. syringae* growth, *lux-1* plants show more chlorosis than WT upon infection at a light intensity of $180 \mu\text{mol m}^{-2} \text{s}^{-1}$. Such increased senescence in *P. syringae*-infected *lux-1* could complicate the manifestation of defense responses of the plant at this light intensity.

Although both LL and DD have been used as free running conditions to test clock activities in different organisms, including plants, animals, and fungi, we recognize that $10 \mu\text{mol m}^{-2} \text{s}^{-1}$ photon flux used in our pathogen assays is a relatively low light intensity, compared with the conditions typically used for plant growth. Nonetheless, such light intensities are encountered in deeply shaded conditions and every day during twilight after dawn and prior to dusk and therefore are physiologically relevant.

Our choice of this light intensity ($10 \mu\text{mol m}^{-2} \text{s}^{-1}$ photon flux) for the pathogen assays reported here was quite serendipitous. As we had reported in the first submission that with $180 \mu\text{mol m}^{-2} \text{s}^{-1}$ photon flux density, we saw enhanced disease susceptibility in the *lux-1* mutant to *P. syringae* spray-infection in LD but not in LL (now Figure S2). With disappointment, the students left the infected plants in the lab. After a few days (when the lucky plants had not been tossed

away), we actually saw more severe disease symptoms in *lux* mutants with both *P. syringae* and Botrytis infection. These observations prompted us to lower the light intensity to the intensity of room light (which in this case was $10 \mu\text{mol m}^{-2} \text{s}^{-1}$ photon flux density) as the LL condition for our pathogen infections.

In addition to light intensity, other factors such as light duration and temperature contribute to LUX-regulated processes. For instance, the early flowering phenotype conferred by the *lux* mutants is more evident in 8 h L/16 h D than in 16 h L/8 h D⁴. The transcriptional targets of LUX (and its interactor ELF3) are temperature-dependent, suggesting a temperature input to EC function^{2,5}. Together, these observations suggest the complexity of the circadian clock regulation of biological processes, which can be further compounded by additional factors that modulate the process either directly or indirectly via an effect on the clock. Further studies that systematically investigate how light, temperature, and other environmental factors interact with the circadian clock to affect biological processes, including pathogen responses, should be worthwhile.

Looking at differentially expressed genes a number of LUX regulated genes was uncovered. Given that in these experiments sampling was done at only two time points (ZT1 and ZT13) and that most LUX targets exhibit daily rhythms, it is imperative to address if the phase of expression of rhythmic genes is the same in all genetic backgrounds (wt, *lux*, *acd6* and *lux/acd6*) used in the experiment (different phases in each background would mislead the identification of differentially expressed genes).

It is important to note that although known as an arrhythmic in LL, *lux-1* shows robust driven rhythms in gene expression in LD that, at least for the luciferase (*LUC*) reporter driven by the *CAB2* or *GRP7* (also called *CCR2*^{1,6}) promoter (*CAB2:LUC* or *GRP7:LUC*), is indistinguishable from that in WT seedlings in terms of period, amplitude, and phase⁴. We confirmed this rhythmic gene expression in *lux-1* by qRT-PCR (new Figure S6). We found that *PRR9*, *PRR7*, *PRR5*, and *LUX* showed distinct expression peaks in Col-0, which are similar to those in *lux-1*. Expression of these genes was higher in *lux-1* than in Col-0 at each time point tested, consistent with the repressor function of LUX in regulating expression of these genes in LD. However, because LUX cycles in abundance, it is possible that at some time points the relief of repression in *lux-1* is indirect, mediated via relief of repression by other transcriptional repressors that are less abundant or less active in the *lux-1* mutant, or via increased transcriptional activation via activators that are more abundant or active in the *lux-1* mutant. *acd6-1* does not affect clock activity¹. Therefore, we believe that the altered expression of cycling genes in LD caused by *lux-1* is unlikely to be due to altered circadian phase among Col-0, *lux-1*, *acd6-1*, and *acd6-1lux-1*. Nevertheless, because there were only two time points (ZT1 and ZT13) used in our RNAseq analysis, we may have missed some cycling genes that are affected by LUX at other times of day.

In addition, as I mentioned before follow up experiments (i.e. using genetics) are needed to support the role of identified genes, such as EDS1 and JAZ5, in mediating LUX regulation of defense responses.

As shown in our previous responses to reviewers' comments, we believe that we already

presented data to support the role of LUX target, EDS1, in mediating LUX regulation of defense response.

Briefly, *EDS1* is a major SA regulator involved in the SA-signal amplification loop^{7,8}. Loss of function in both *EDS1* and *LUX* leads to similar phenotypes, including suppression of *acd6-1*-conferred phenotypes, reduced SA accumulation, and enhanced *P. syringae* susceptibility (Figure 1, 2B, and^{8,9}). Therefore, *LUX* could act through *EDS1* in regulating SA-mediated defense. The suppressed expression of some other SA genes and SA accumulation in *lux-1* could be due to reduced *EDS1* expression in *lux-1*. We further provide data on the role of EDS1 in other LUX-mediated defense phenotypes, including *Botrytis* response, JA sensitivity, and JA signaling under pathogen challenge (Figures 2C, 4A, and 4C). These data reveal a JA-regulatory function of EDS1. Interestingly, the *eds1-2* mutant was not compromised in *Botrytis* resistance (Figure 2C). Therefore, we conclude that *LUX* only acts partially through *EDS1* to affect SA signaling and/or JA signaling.

We agree with this reviewer that genetic analysis, e.g. introducing *eds1-2* into *lux-1*, could be a follow-up experiment that may help further clarify some aspects of EDS1 function in LUX-mediated defense. We have begun to generate these additional *LUX* and *EDS1* related genetic materials. Given the time required to generate these mutants, we think that characterization of these plant materials is beyond the scope of this report.

JAZ5 is a gene from a large gene family and the single mutants of the family members mostly do not show defense defects. Therefore, it would be difficult to use a loss of function approach, as we did with *EDS1*, to study the role of *JAZ5* in *LUX*-mediated defense. However, we do intend to follow up with *JAZ5*, using other approaches, such as gain of function approaches and multiple JAZ knockouts, to assess *JAZ5* and its homologs in LUX-mediated defense in the future.

Finally, I find that experiments that attempted to establish clock regulation by JA, MJ or JA-Ile should be further revised. First, I find intriguing that the period length under LL (10uE) was close to 24h. As previously shown, under such low light intensity the period should have been significantly longer.

We appreciate very much this point raised by the reviewer. In the previous version, we mistakenly reported the light intensity of our clock assay room ($10 \mu\text{mol m}^{-2} \text{s}^{-1}$ photon flux) as the light intensity for our clock assays. Our clock assay system actually includes LED light panels on both open sides of the plate stackers that illuminate 96-well plates during the whole recording period for each experiment. The light intensity on each 96-well plate is about $90 \mu\text{mol m}^{-2} \text{s}^{-1}$ photon flux. This value takes into the consideration of microplates being stacked and reflects the light intensity received by the seedlings. We believe that this light intensity is similar to or greater than intensities used by many people for clock assays with plant seedlings. For example, Michael and McClung¹⁰ and Salomé et al¹¹ reported periods of ~ 24.5 h at fluence rates of $15\text{-}25 \mu\text{mol m}^{-2} \text{s}^{-1}$. A period of about 24 h was reported when the light intensity was $60\text{-}70 \mu\text{mol m}^{-2} \text{s}^{-1}$ ^{4,12,13}. Thus, we would not expect to observe long period at the greater light intensity that we used. We apologize for the confusion caused by our mistake.

Second, period length and phase phenotypes were observed with only the *GRP7:LUC* reporter. While authors indicate that this reporter may be more sensitive to the hormone treatment, this result indicates that either two clocks are functioning simultaneously with a different period or that two different tissue specific clocks can run with a different period. Such result would be highly novel but would require further supporting experiments.

The reviewer is correct that with MJ treatment, we observed period lengthening and phase shift with the *GRP7:LUC* reporter but not with the *CCA1:LUC* reporter. However, as mentioned in the next point, we have replaced the MJ data with JA-Ile data in Figure 5. JA-Ile treatment at the doses employed does not inhibit seedling growth under our clock assay condition. Moreover, JA-Ile treatment significantly lengthens the period of expression of both the *CCA1* and *GRP7* reporters.

Clock reporters showing differential responses to treatments is not new. Examples of such cases can be seen in these papers that show reciprocal regulation of the circadian clock by nutrient status, ROS, and phytohormones^{14, 15, 16}. The mechanisms underlying such differential responses have not been well understood. We speculate that the result with the MJ treatment indicates that the reporters have different sensitivity to the treatment. It could also be as the reviewer mentioned, either two clocks are functioning simultaneously with a different period in one tissue or that two different tissue specific clocks can run with different periods.

Third, the amplitude of *CCA1* and *GRP7* rhythms are consistently reduced in a dose dependent manner. However, JA was reported to significantly reduce plant growth in a dose dependent manner, which could have biased the amplitude phenotype when using a luciferase reporter. Normalization by plant size, or other methods (i.e. gene expression by qPCR) could be considered.

We appreciate this insightful comment from the reviewer. We did observe growth inhibition by MJ in a dosage dependent manner in our clock assays (new Figure S8). With pictures of seedlings on microplates taken after each luminescence recording, we are able to determine plant size by measuring leaf area, using Image J. Relative leaf area of seedlings from different genotype with MJ treatment conducted at subjective morning or subjective evening is presented in this revision as Figure S8. Amplitude of MJ-treated samples are normalized to their corresponding relative leaf area and is presented in Figure S9. Even with this normalization, the general conclusion that MJ dampens greatly clock amplitude remains the same as we reported in the previous version.

On the other hand, JA isoleucine (JA-Ile), a major bioactive JA derivative that binds to COI1 to activate JA signaling¹⁷, did not cause seedling growth inhibition under the same condition used for MJ treatment (Figure S8B), suggesting that these two chemicals act differently to regulate plant growth. Similar to MJ treatment, JA-Ile treatment also induced drastic amplitude dampening with both *CCA1:LUC* and *GRP7:LUC* reporters in Col-0 (Figure 5A, 5B, 5I, and 5J). The period of both reporters in Col-0 was lengthened in the presence of 100 μ M JA-Ile at subjective dawn, suggesting a higher morning-sensitivity of the reporters to JA-Ile treatment (Figure 5C and 5K). We did not observe a phase change of the two reporters with JA-Ile

treatment. To further support the specificity of JA signaling on clock regulation, *CCA1:LUC* expressed in the *coi1-17* mutant did not change in amplitude, period, and phase with JA-Ile treatments (Figure 5E to 5H). We believe that these data support a reciprocal regulation of the circadian clock by JA signaling.

Considering that MJ suppression of plant growth makes it difficult to distinguish the direct effect of MJ on clock activity from the secondary effect due to its growth inhibition and that growth inhibition may also complicate the display of clock-regulated phenotypes in plants, we present the JA-Ile data as Figure 5 and MJ data as Figure S9 in this revision.

Finally, if LUX is a coordinator of clock and defense responses, what would be the role of LUX in mediating JA, MJ or JA-Ile regulation of the clock function?

Our detailed answer to this question is presented in the Discussion section on P14. Briefly, we believe that the role of LUX in regulating JA (and SA) signaling includes but is not limited to the following:

- a. A LUX-mediated circadian clock continuously monitors the change of JA and SA signaling to ensure proper growth, development, and response to external stimuli.
- b. The reciprocal regulation of LUX-circadian clock by JA signaling provides another layer of monitoring of defense signaling pathways, which can be reset by their own feedback inhibition of the circadian clock.
- c. The fact that LUX regulates JA signaling and its own expression is also influenced by JA clearly suggests LUX is a key node in mediating crosstalk between the circadian clock and defense signal involving JA. We also think besides LUX, other clock genes are likely involved in clock-defense crosstalk through SA, JA, and/or other defense signaling pathways.

Reviewer #4 (Remarks to the Author):

The authors demonstrated comprehensive studies to understand molecular mechanisms underlying relationship between circadian clock and innate immunity in Arabidopsis. They found that *lux* mutants suppressed constitutive defense phenotypes of *acd6*. Phenotypic and genome-wide gene expression analyses suggested possible molecular mechanism of LUX for innate immunity through ACD6. They showed that strong antagonist of JA signaling can modulate circadian clock.

It was clear that LUX is involved in defense response, since the authors further analyzed two independent *lux* alleles and *elf3* to confirm that LUX (and ELF3, an interaction of LUX) is involved in defense response. CHIP-qPCR experiment with appropriate control loci confirmed that LUX associates with *JAZ5* and *EDS* promoters.

However, I have still concern about the conclusion that JA signaling controls the clock, which seems to be proposed by Fig. 5 and Supplemental Fig. 6 and 7. I appreciated the authors effort to consider if JA signaling controls the clock. However, the data presented here were not convincing to support their propose. The authors found that amplitude of both morning and evening reporters (*CCA1:LUC* and *GPR7:LUC*) were decreased, but period length were not changed. This suggests that overall plant vigorousness or activity was decreased upon JA

treatment, but does not suggest circadian clock is controlled by JA signaling. Again, even though such decreased amplitude, I see that most crucial parameter of the clock, period length, were not drastically changed, suggesting circadian clock is robust against JA treatment.

We appreciate this reviewer's recognition of the importance of our work in advancing the molecular mechanisms underlying crosstalk between circadian clock and innate immunity in *Arabidopsis*. Regarding this reviewer's concern about the conclusion that JA signaling controls the clock, we have addressed this concern in our response to the last points of Reviewer #3.

References

1. Zhang C, *et al.* Crosstalk between the circadian clock and innate immunity in *Arabidopsis*. *PLoS Pathog* **9**, e1003370 (2013).
2. Ezer D, *et al.* The evening complex coordinates environmental and endogenous signals in *Arabidopsis*. *Nat Plants* **3**, 17087 (2017).
3. Zhang Y, *et al.* Circadian evening complex represses jasmonate-induced leaf senescence in *Arabidopsis*. *Mol Plant* **11**, 326-337 (2018).
4. Hazen SP, Schultz TF, Pruneda-Paz JL, Borevitz JO, Ecker JR, Kay SA. LUX ARRHYTHMO encodes a Myb domain protein essential for circadian rhythms. *Proc Natl Acad Sci U S A* **102**, 10387-10392 (2005).
5. Box MS, *et al.* ELF3 controls thermoresponsive growth in *Arabidopsis*. *Curr Biol* **25**, 194-199 (2015).
6. Carpenter CD, Kreps JA, Simon AE. Genes encoding glycine-rich *Arabidopsis thaliana* proteins with RNA-binding motifs are influenced by cold treatment and an endogenous circadian rhythm. *Plant Physiol* **104**, 1015-1025 (1994).
7. Bartsch M, *et al.* Salicylic acid-independent ENHANCED DISEASE SUSCEPTIBILITY1 signaling in *Arabidopsis* immunity and cell death is regulated by the monooxygenase FMO1 and the Nudix hydrolase NUDT7. *Plant Cell* **18**, 1038-1051 (2006).
8. Ng G, Seabolt S, Zhang C, Salimian S, Watkins TA, Lu H. Genetic dissection of salicylic acid-mediated defense signaling networks in *Arabidopsis*. *Genetics* **189**, 851-859 (2011).
9. Falk A, Feys BJ, Frost LN, Jones JD, Daniels MJ, Parker JE. EDS1, an essential component of R gene-mediated disease resistance in *Arabidopsis* has homology to eukaryotic lipases. *Proc Natl Acad Sci U S A* **96**, 3292-3297. (1999).
10. Michael TP, McClung CR. Phase-specific circadian clock regulatory elements in *Arabidopsis*. *Plant Physiol* **130**, 627-638 (2002).

11. Salome PA, McClung CR. *PSEUDO-RESPONSE REGULATOR 7 and 9 are partially redundant genes essential for the temperature responsiveness of the Arabidopsis circadian clock. Plant Cell* **17**, 791-803 (2005).
12. Li Z, Bonaldi K, Uribe F, Pruneda-Paz JL. A localized *Pseudomonas syringae* infection triggers systemic clock responses in Arabidopsis. *Curr Biol* **28**, 630-639 e634 (2018).
13. Xie Q, *et al.* LNK1 and LNK2 are transcriptional coactivators in the Arabidopsis circadian oscillator. *Plant Cell*, (2014).
14. Lai AG, Doherty CJ, Mueller-Roeber B, Kay SA, Schippers JH, Dijkwel PP. CIRCADIAN CLOCK-ASSOCIATED 1 regulates ROS homeostasis and oxidative stress responses. *Proc Natl Acad Sci U S A* **109**, 17129-17134 (2012).
15. Hanano S, Domagalska MA, Nagy F, Davis SJ. Multiple phytohormones influence distinct parameters of the plant circadian clock. *Genes Cells* **11**, 1381-1392 (2006).
16. Hong S, Kim SA, Guerinot ML, McClung CR. Reciprocal interaction of the circadian clock with the iron homeostasis network in Arabidopsis. *Plant Physiol* **161**, 893-903 (2013).
17. Koo AJ, Howe GA. Catabolism and deactivation of the lipid-derived hormone jasmonoyl-isoleucine. *Front Plant Sci* **3**, 19 (2012).

Responses to Reviewers' Comments:

Reviewer #1 (Remarks to the Author):

The revised manuscript by Zhang et al. has addressed most of my previous concerns. There are still a few points left:

1. Lines 94-96: As the authors also mentioned in the discussion part, Psm avrRpt2 induced systemic SA level increase has been shown to be dependent on CHE, a clock gene. Therefore it is not appropriate here to say “no clock genes have yet been reported so far to play such a role in SA regulation”

Here we meant to say that no clock genes have yet been reported so far to play such a role in regulating "acute SA accumulation in the local infected region". We clarified this point in the revised text. CHE was only shown to affect SA levels in the SAR tissue, which are much lower than those induced in the local infected tissue. CHE has not been demonstrated to affect local SA accumulation with *P. syringae* infection.

2. Fig. S3B: missing connectors linking Col-0.1, Col-0.2, Col-0.3

Connectors linking Col-0.1, Col-0.2, Col-0.3 are shown now.

3. Line 234: base on the GO analysis, it appears that the clustering analysis did not generate much differences in the three groups.

There are some differences among the three groups as we clarify in the text (also see below).

"Cluster analysis of the 1618 *LUX*-affected genes revealed three major groups (Figure 3A). Expression of many genes in group II was relatively low in all four genotypes, compared with those in groups I and III. Some genes in group II showed greater expression in *lux-1*, supporting a known role of *LUX* as a transcriptional repressor. Most genes in group I were highly induced in *acd6-1*. Expression of most group I and III genes was suppressed by *lux-1*, especially at ZT1 and/or in the *acd6-1* background, suggesting that *LUX* can also positively affect gene expression via direct or indirect means. GO analysis revealed that groups I and III were more enriched than group II in genes responding to abiotic and biotic stimuli (Figure 3A)."

4. Line 236: “and II” should be “and III”?

This is corrected now.

5. Line 240: group II is also enriched in genes responding to abiotic and biotic stimuli

We modified the text to reflect this point as the follow:

"GO analysis revealed that groups I and III were more enriched than group II in genes responding to abiotic and biotic stimuli."

6. Lines 249-250: the qPCR results seem do not support statistically significant changes of MYC2

There is no statistical significance in MYC2 expression among the four genotypes. We made the correction. We appreciate the reviewer noticing our error.

7. Lines 272-276: it is quite strange that the authors used Pma instead of Botrytis to study the response of JA marker genes since Pma is usually used as a model of biotrophic pathogen while JA is more involved in defence against necrotrophic pathogens.

The point of this paragraph and the related Figure 4B and 4C is to show that LUX and EDS1 could affect JA signaling under defense conditions. Both Pma and Botrytis can activate host defense and are known to affect JA signaling during infection. We did observe the effect of LUX and EDS1 on JA gene expression with Pma infection (Figure 4B and 4C).

The reason that we prefer using Pma rather than Botrytis for a pathogen-induced gene expression study is that Pma-infected plants are kept in the same growth condition used for plant growth. There is no need to cover the plants. In the case of Botrytis infection, the fungus needs high humidity to infect plants well and, therefore, Botrytis-infected plants are covered during the infection process. We feel that high humidity may complicate gene expression results in some genetic backgrounds. Nevertheless, we agree with this reviewer that gene expression analysis with Botrytis infection has been done and is an alternative way to gauge the changes in expression of JA genes and other defense genes.

8. Reference number 1 and 17 are the same paper.

We corrected this mistake.

Reviewer #2 (Remarks to the Author):

You have addressed my concerns, and I am satisfied with the revisions

Reviewer #3 (Remarks to the Author):

In this revised version, authors included additional experiments and reformatted/edited many sections of the manuscript. While I appreciate authors' efforts to improve the manuscript I still think that this work does not establish a role for LUX as a "coordinator of clock and defense responses". As I mentioned before, most experiments were performed in LD, suggesting a role for LUX in regulating defenses in light/dark cycles. In fact LUX (and the EC) was shown to regulate light signaling. New infection experiments in LL seem to indicate that this is the case, as defense phenotypes were only observed when plants were grown under a very low light intensity (typically not used in clock experiments). Furthermore, experiments in figure 2B indicate that lux mutant plants are more resistant to infections at ZT13 compared to ZT1, indicating that clock regulation of defenses against *P. syri* was not affected in the lux mutant background. Increased

susceptibility to the pathogen was observed after both ZT1 and ZT13 infections suggesting that the overall defense was affected.

We appreciate this reviewer raised these insightful points.

1) We show in this report that *LUX* loss of function plants are more susceptible to *P. syringae* and *Botrytis* in a free running condition (LL with $10 \mu\text{mol m}^{-2} \text{s}^{-1}$ photon flux density) (Figure 2B, 2C, and 4D) and that resistance is rescued by the functional *LUX-GFP* gene. Thus we provide critical data to demonstrate a circadian regulation of plant defense by LUX.

2) We had also been puzzled by the fact that the arrhythmic clock mutant *lux* keeps temporal difference in resistance to *P. syringae* spray-infected in the morning and at night, as shown in Col-0 (Figure 2B). We performed additional experiments in LL in order to seek reasons behind this interesting phenotype displayed by *lux*. Our results support a role of LUX in regulating temporal defense to both infiltration and spray-infected *P. syringae*. The difference in *P. syringae* growth between morning- and evening-spray infected *lux* is due to the disruption of temporal stomatal behavior, the underlying mechanism of defense against spray-infected bacteria. Here we summarize our results regarding reviewer's points raised here as the following.

a) Plants are known to employ different mechanisms to fight against pathogens and pests with different lifestyles at different times of day. For infiltrated *P. syringae*, plants with a normal circadian clock show higher susceptibility at night than in the morning (Figure 2C and ^{1,2,3}; these are new data). Such time-dependent susceptibility was abolished in *lux*, which demonstrated similar *PmaDG3* growth when infected at both LL25 and LL37 (Figure 2C). More bacterial growth was found in the *lux* mutants than Col-0 with *PmaDG3* infection at LL25. Thus LUX influences circadian-regulated defense against infiltrated *P. syringae*, resistance to which requires stomata-independent pathway.

b) To spray-infected *PmaDG3*, Col-0 showed more susceptibility at LL25 than at LL37 (Figure 2B). Interestingly, while they showed greater bacterial growth than Col-0 when infected at both times, the two *lux* mutants also demonstrated higher sensitivity to *PmaDG3* in the morning than at night (Figure 2B). We repeated these experiments and also tested additional clock mutants (manuscript in preparation). We obtained similar results from these experiments.

c) These observations appear to suggest that the circadian clock does not contribute to defense against epiphytic bacteria (spray-infected bacteria). However, our further analysis of the underlying mechanism of defense against spray-infected *P. syringae*, the change of stomatal aperture, rejected this notion. Our data show:

In LL and in the absence of *P. syringae*, Col-0 showed circadian-regulated stomatal aperture, being more open in the morning than at night (Figure 2D and S2B; these are new data). Consistent with they being arrhythmic, we found that the *lux* mutants lost such a temporal stomatal activity, keeping stomata open at both LL25 and LL37 (Figure 2D and S2B).

In LL and in the presence of *P. syringae*, stomata of Col-0 were highly sensitive for aperture reduction in the morning but showed no response at night. In contrast, the *lux* mutants lost this

temporal gating of the response to acute *P. syringae* infection, showing stomatal aperture reduction both in the morning and at night. Open stomata observed in the *lux* mutants at LL37 could allow *PmaDG3* to enter into plant tissue to cause infection, making the *lux* plants more susceptible than Col-0 to *PmaDG3* infected at LL37. The higher sensitivity of *lux* stomata to acute *PmaDG3* infection at night than in the morning explains why the *lux* mutants showed more resistance at night than in the morning.

We believe that these data provide another mechanism of circadian regulation of plant defense by LUX, which is through gating stomata aperture under free running and acute *P. syringae* challenge conditions. The multiple functions of LUX in regulating physical barrier posed by stomata and defense signaling mediated by SA and JA underscore the importance of the circadian clock gene *LUX* in broad disease resistance.

3) While establishing that the core clock gene *LUX* regulates plant innate immunity, enhanced disease susceptibility of the *lux* mutants is shown in LL with light intensity of $10 \mu\text{mol m}^{-2} \text{s}^{-1}$ photon flux but not of $180 \mu\text{mol m}^{-2} \text{s}^{-1}$ photon flux. These results suggest that the defense role of the circadian clock is conditional and influenced by light. The EC, consisting of LUX-ELF3-ELF4, is known to regulate light signaling through affecting expression of many photosynthesis genes and light response genes⁴. Thus, mutations in *LUX* or other EC genes could make plants particularly sensitive to light. Consistent with this, *LUX* was shown to be a repressor of leaf senescence⁵. Although not supporting higher *P. syringae* growth, *lux-1* plants show more chlorosis than WT upon infection at a light intensity of $180 \mu\text{mol m}^{-2} \text{s}^{-1}$. Such increased senescence in *P. syringae*-infected *lux-1* could complicate the manifestation of defense responses of the plant at this light intensity.

Our use of a lower light intensity, $10 \mu\text{mol m}^{-2} \text{s}^{-1}$ photon flux, allowed a detection of the difference in pathogen responses between Col-0 and *lux* plants. We recognize though that this is a relatively low light intensity, compared with the conditions typically used for plant growth in the laboratory. Nonetheless, such light intensities are encountered in deeply shaded conditions and every day during twilight after dawn and prior to dusk. In addition, both LL and DD have been routinely used as free running conditions to test clock activities in different organisms, including plants, animals, and fungi, in laboratory conditions. Therefore, our use of this low light regime is physiologically relevant.

In addition to light intensity, other factors such as light duration and temperature contribute to LUX-regulated processes. For instance, the early flowering phenotype conferred by the *lux* mutants is more evident in 8 h L/16 h D than in 16 h L/8 h D⁶. The transcriptional targets of LUX (and its interactor ELF3) are temperature-dependent, suggesting a temperature input to EC function^{4,7}. Together, these observations suggest the complexity of the circadian clock regulation of biological processes, which can be further compounded by additional factors that modulate the process either directly or indirectly via an effect on the clock. Further studies that systematically investigate how light, temperature, and other environmental factors interact with the circadian clock to affect biological processes, including pathogen responses, would be worthwhile.

Looking at differentially expressed genes a number of LUX regulated genes was uncovered. Given that in these experiments sampling was done at only two time points (ZT1 and ZT13) and that most LUX targets exhibit daily rhythms, it is imperative to address if the phase of expression of rhythmic genes is the same in all genetic backgrounds (wt, *lux*, *acd6* and *lux/acd6*) used in the experiment (different phases in each background would mislead the identification of differentially expressed genes).

It is important to note that although known as an arrhythmic in LL, *lux-1* shows robust driven rhythms in gene expression in LD that, at least for the luciferase (*LUC*) reporter driven by the *CAB2* or *GRP7* (also called *CCR2*^{1,8}) promoter (*CAB2:LUC* or *GRP7:LUC*), is indistinguishable from that in WT seedlings in terms of period, amplitude, and phase⁶. We confirmed this rhythmic gene expression in *lux-1* by qRT-PCR (new Figure S6). We found that *PRR9*, *PRR7*, *PRR5*, and *LUX* showed distinct expression peaks in Col-0, which are similar to those in *lux-1*. Expression of these genes was higher in *lux-1* than in Col-0 at each time point tested, consistent with the repressor function of LUX in regulating expression of these genes in LD. However, because LUX cycles in abundance, it is possible that at some time points the relief of repression in *lux-1* is indirect, mediated via relief of repression by other transcriptional repressors that are less abundant or less active in the *lux-1* mutant, or via increased transcriptional activation via activators that are more abundant or active in the *lux-1* mutant. *acd6-1* does not affect clock activity¹. Therefore, we believe that the altered expression of cycling genes in LD caused by *lux-1* is unlikely to be due to altered circadian phase among Col-0, *lux-1*, *acd6-1*, and *acd6-1lux-1*. Nevertheless, because there were only two time points (ZT1 and ZT13) used in our RNAseq analysis, we may have missed some cycling genes that are affected by *LUX* at other times of day.

In addition, as I mentioned before follow up experiments (i.e. using genetics) are needed to support the role of identified genes, such as *EDS1* and *JAZ5*, in mediating LUX regulation of defense responses.

As shown in our previous responses to reviewers' comments, we believe that we already presented data to support the role of LUX target, *EDS1*, in mediating LUX regulation of defense response.

Briefly, *EDS1* is a major SA regulator involved in the SA-signal amplification loop^{9,10}. Loss of function in both *EDS1* and *LUX* leads to similar phenotypes, including suppression of *acd6-1*-conferred phenotypes, reduced SA accumulation, and enhanced *P. syringae* susceptibility (Figure 1, 2B, and^{10,11}). Therefore, *LUX* could act through *EDS1* in regulating SA-mediated defense. The suppressed expression of some other SA genes and SA accumulation in *lux-1* could be due to reduced *EDS1* expression in *lux-1*. We further provide data on the role of *EDS1* in other LUX-mediated defense phenotypes, including *Botrytis* response, JA sensitivity, and JA signaling under pathogen challenge (Figures 4A, 4C, and 4D). These data reveal a JA-regulatory function of *EDS1*. Interestingly, the *eds1-2* mutant was not compromised in *Botrytis* resistance (Figure 4D). Therefore, we conclude that *LUX* only acts partially through *EDS1* to affect SA signaling and/or JA signaling.

We agree with this reviewer that genetic analysis, e.g. introducing *eds1-2* into *lux-1*, could be a

follow-up experiment that may help further clarify some aspects of EDS1 function in LUX-mediated defense. We have begun to generate these additional *LUX* and *EDS1* related genetic materials. Given the time required to generate these mutants, we think that characterization of these plant materials is beyond the scope of this report.

JAZ5 is a gene from a large gene family and the single mutants of the family members mostly do not show defense defects. Therefore, it would be difficult to use a loss of function approach, as we did with *EDS1*, to study the role of *JAZ5* in *LUX*-mediated defense. However, we do intend to follow up with *JAZ5*, using other approaches, such as gain of function approaches and multiple *JAZ* knockouts, to assess *JAZ5* and its homologs in *LUX*-mediated defense in the future.

Finally, I find that experiments that attempted to establish clock regulation by JA, MJ or JA-Ile should be further revised. First, I find intriguing that the period length under LL (10uE) was close to 24h. As previously shown, under such low light intensity the period should have been significantly longer.

We appreciate very much this point raised by the reviewer. In the previous version, we mistakenly reported the light intensity of our clock assay room ($10 \mu\text{mol m}^{-2} \text{s}^{-1}$ photon flux) as the light intensity for our clock assays. Our clock assay system actually includes LED light panels on both open sides of the plate stackers that illuminate 96-well plates during the whole recording period for each experiment. The light intensity on each 96-well plate is about $90 \mu\text{mol m}^{-2} \text{s}^{-1}$ photon flux. We believe that this light intensity is similar to or greater than intensities used by many people for clock assays with plant seedlings. For example, Michael and McClung¹² and Salomé et al¹³ reported periods of ~24.5 h at fluence rates of 15-25 $\mu\text{mol m}^{-2} \text{s}^{-1}$. A period of about 24 h was reported when the light intensity was 60-70 $\mu\text{mol m}^{-2} \text{s}^{-1}$ ^{6, 14, 15}. Thus, we would not expect to observe longer period at the greater light intensity that we used. We apologize for the confusion caused by our mistake.

Second, period length and phase phenotypes were observed with only the *GRP7:LUC* reporter. While authors indicate that the this reporter may be more sensitive to the hormone treatment, this result indicates that either two clocks are functioning simultaneously with a different period or that to different tissue specific clocks can run with a different period. Such result would be highly novel but would require of further supporting experiments.

The reviewer is correct that with MJ treatment, we observed period lengthening and phase shift with the *GRP7:LUC* reporter but not with the *CCA1:LUC* reporter. MJ treatment suppressed seedling growth during the clock assay, making it difficult to disentangle the direct effect on the circadian clock by MJ-induced signaling from indirect effects caused by plant growth inhibition. Unlike MJ, JA-Ile treatment at the doses employed does not inhibit seedling growth under our clock assay condition. Moreover, JA-Ile treatment significantly lengthens the period of expression of both the *CCA1* and *GRP7* reporters in addition to damping clock amplitude. Thus, we have replaced the MJ data with JA-Ile data in Figure 5 in this revision. We further addressed this point in our response to the next comment made by this reviewer.

Clock reporters showing differential responses to treatments is not new. Examples of such cases can be seen in these papers that show reciprocal regulation of the circadian clock by nutrient status, ROS, and phytohormones^{16, 17, 18}. The mechanisms underlying such differential responses have not been well understood. We speculate that the result with the MJ treatment indicates that the reporters have different sensitivity to the treatment. It could also be as the reviewer mentioned, either two clocks are functioning simultaneously with a different period in one tissue or that two different tissue specific clocks can run with different periods.

Third, the amplitude of *CCA1* and *GRP7* rhythms are consistently reduced in a dose dependent manner. However, JA was reported to significantly reduce plant growth in a dose dependent manner, which could have biased the amplitude phenotype when using a luciferase reporter. Normalization by plant size, or other methods (i.e. gene expression by qPCR) could be considered.

We appreciate this insightful comment from the reviewer. We did observe growth inhibition by MJ in a dosage dependent manner in our clock assays (new Figure S8). With pictures of seedlings on microplates taken after each luminescence recording, we are able to determine plant size by measuring leaf area, using Image J. Relative leaf area of seedlings from different genotype with MJ treatment conducted at subjective morning or subjective evening is presented in this revision as Figure S8. Amplitude of MJ-treated samples are normalized to their corresponding relative leaf area and is presented in Figure S9. Even with this normalization, the general conclusion that MJ dampens greatly clock amplitude remains the same as we reported in the previous version.

Unlike MJ, JA isoleucine (JA-Ile), a major bioactive JA derivative that binds to COI1 to activate JA signaling¹⁹, did not cause seedling growth inhibition under the same condition used for MJ treatment (Figure S8B), suggesting that these two chemicals act differently to regulate plant growth. Similar to MJ treatment, JA-Ile treatment also induced drastic amplitude dampening with both *CCA1:LUC* and *GRP7:LUC* reporters in Col-0 (Figure 5A, 5B, 5I, and 5J). The period of both reporters in Col-0 was lengthened in the presence of 100 μ M JA-Ile at subjective dawn, suggesting a higher morning-sensitivity of the reporters to JA-Ile treatment (Figure 5C and 5K). We did not observe a phase change of the two reporters with JA-Ile treatment. To further support the specificity of JA signaling on clock regulation, *CCA1:LUC* expressed in the *coi1-17* mutant did not change in amplitude, period, and phase with JA-Ile treatments (Figure 5E to 5H). We believe that these data support a reciprocal regulation of the circadian clock by JA signaling.

Considering that MJ suppression of plant growth makes it difficult to distinguish the direct effect of MJ on clock activity from the secondary effect due to its growth inhibition and that growth inhibition may also complicate the display of clock-regulated phenotypes in plants, we present the JA-Ile data as Figure 5 and MJ data as Figure S9 in this revision.

Finally, if LUX is a coordinator of clock and defense responses, what would be the role of LUX in mediating JA, MJ or JA-Ile regulation of the clock function?

Our detailed answer to this question is presented in the Discussion section on P15-16. Briefly, we believe that the role of LUX in regulating JA (and SA) signaling includes but is not limited to the following:

- a. A LUX-mediated circadian clock continuously monitors the change of JA and SA signaling to ensure proper growth, development, and response to external stimuli.
- b. The reciprocal regulation of LUX-circadian clock by JA signaling provides another layer of monitoring of defense signaling pathways, which can be reset by their own feedback inhibition of the circadian clock.
- c. The fact that LUX regulates JA signaling and its own expression is also influenced by JA clearly suggests LUX is a key node in mediating crosstalk between the circadian clock and defense signal involving JA. We also think besides LUX, other clock genes are likely involved in clock-defense crosstalk through SA, JA, and/or other defense signaling pathways.

Reviewer #4 (Remarks to the Author):

The authors demonstrated comprehensive studies to understand molecular mechanisms underlying relationship between circadian clock and innate immunity in Arabidopsis. They found that lux mutants suppressed constitutive defense phenotypes of *acd6*. Phenotypic and genome-wide gene expression analyses suggested possible molecular mechanism of LUX for innate immunity through *ACD6*. They showed that strong antagonist of JA signaling can modulate circadian clock.

It was clear that LUX is involved in defense response, since the authors further analyzed two independent lux alleles and *elf3* to confirm that LUX (and *ELF3*, an interaction of LUX) is involved in defense response. ChIP-qPCR experiment with appropriate control loci confirmed that LUX associates with *JAZ5* and *EDS* promoters.

However, I have still concern about the conclusion that JA signaling controls the clock, which seems to be proposed by Fig. 5 and Supplemental Fig. 6 and 7. I appreciated the authors effort to consider if JA signaling controls the clock. However, the data presented here were not convincing to support their propose. The authors found that amplitude of both morning and evening reporters (*CCA1:LUC* and *GPR7:LUC*) were decreased, but period length were not changed. This suggests that overall plant vigorousness or activity was decreased upon JA treatment, but does not suggest circadian clock is controlled by JA signaling. Again, even though such decreased amplitude, I see that most crucial parameter of the clock, period length, were not drastically changed, suggesting circadian clock is robust against to JA treatment.

We appreciate this reviewer's recognition of the importance of our work in advancing the molecular mechanisms underlying crosstalk between circadian clock and innate immunity in Arabidopsis. Regarding this reviewer's comment on the conclusion that JA signaling controls the clock, we have addressed this comment in our response to Reviewer #3. Briefly, our data show that JA isoleucine (JA-Ile), a major bioactive JA, strongly effects clock amplitude and lengthens clock period. Unlike MJ, JA does not negatively affect plant growth. These results also suggest that 1) different JA agonists have differential effects on plant growth and clock activity; and 2) different clock gene reporters may have differential response to a certain treatment. For detail, please see our responses to the second and third points raised by Reviewer #3.

References

1. Zhang C, *et al.* Crosstalk between the circadian clock and innate immunity in Arabidopsis. *PLoS Pathog* **9**, e1003370 (2013).
2. Bhardwaj V, Meier S, Petersen LN, Ingle RA, Roden LC. Defence responses of *Arabidopsis thaliana* to infection by *Pseudomonas syringae* are regulated by the circadian clock. *PLoS One* **6**, e26968 (2011).
3. Shin J, Heidrich K, Sanchez-Villarreal A, Parker JE, Davis SJ. TIME FOR COFFEE represses accumulation of the MYC2 transcription factor to provide time-of-day regulation of jasmonate signaling in Arabidopsis. *Plant Cell* **24**, 2470-2482 (2012).
4. Ezer D, *et al.* The evening complex coordinates environmental and endogenous signals in Arabidopsis. *Nat Plants* **3**, 17087 (2017).
5. Zhang Y, *et al.* Circadian evening complex represses jasmonate-induced leaf senescence in Arabidopsis. *Mol Plant* **11**, 326-337 (2018).
6. Hazen SP, Schultz TF, Pruneda-Paz JL, Borevitz JO, Ecker JR, Kay SA. LUX ARRHYTHMO encodes a Myb domain protein essential for circadian rhythms. *Proc Natl Acad Sci U S A* **102**, 10387-10392 (2005).
7. Box MS, *et al.* ELF3 controls thermoresponsive growth in Arabidopsis. *Curr Biol* **25**, 194-199 (2015).
8. Carpenter CD, Kreps JA, Simon AE. Genes encoding glycine-rich *Arabidopsis thaliana* proteins with RNA-binding motifs are influenced by cold treatment and an endogenous circadian rhythm. *Plant Physiol* **104**, 1015-1025 (1994).
9. Bartsch M, *et al.* Salicylic acid-independent ENHANCED DISEASE SUSCEPTIBILITY1 signaling in Arabidopsis immunity and cell death is regulated by the monooxygenase FMO1 and the Nudix hydrolase NUDT7. *Plant Cell* **18**, 1038-1051 (2006).
10. Ng G, Seabolt S, Zhang C, Salimian S, Watkins TA, Lu H. Genetic dissection of salicylic acid-mediated defense signaling networks in Arabidopsis. *Genetics* **189**, 851-859 (2011).
11. Falk A, Feys BJ, Frost LN, Jones JD, Daniels MJ, Parker JE. EDS1, an essential component of R gene-mediated disease resistance in Arabidopsis has homology to eukaryotic lipases. *Proc Natl Acad Sci U S A* **96**, 3292-3297. (1999).
12. Michael TP, McClung CR. Phase-specific circadian clock regulatory elements in Arabidopsis. *Plant Physiol* **130**, 627-638 (2002).

13. Salome PA, McClung CR. *PSEUDO-RESPONSE REGULATOR 7 and 9* are partially redundant genes essential for the temperature responsiveness of the Arabidopsis circadian clock. *Plant Cell* **17**, 791-803 (2005).
14. Li Z, Bonaldi K, Uribe F, Pruneda-Paz JL. A localized *Pseudomonas syringae* infection triggers systemic clock responses in Arabidopsis. *Curr Biol* **28**, 630-639 e634 (2018).
15. Xie Q, *et al.* LNK1 and LNK2 are transcriptional coactivators in the Arabidopsis circadian oscillator. *Plant Cell*, (2014).
16. Lai AG, Doherty CJ, Mueller-Roeber B, Kay SA, Schippers JH, Dijkwel PP. CIRCADIAN CLOCK-ASSOCIATED 1 regulates ROS homeostasis and oxidative stress responses. *Proc Natl Acad Sci U S A* **109**, 17129-17134 (2012).
17. Hanano S, Domagalska MA, Nagy F, Davis SJ. Multiple phytohormones influence distinct parameters of the plant circadian clock. *Genes Cells* **11**, 1381-1392 (2006).
18. Hong S, Kim SA, Guerinot ML, McClung CR. Reciprocal interaction of the circadian clock with the iron homeostasis network in Arabidopsis. *Plant Physiol* **161**, 893-903 (2013).
19. Koo AJ, Howe GA. Catabolism and deactivation of the lipid-derived hormone jasmonoyl-isoleucine. *Front Plant Sci* **3**, 19 (2012).

REVIEWERS' COMMENTS:

Reviewer #1 (Remarks to the Author):

The revised manuscript by Zhang et al. has addressed my previous concerns.

Reviewer #3 (Remarks to the Author):

In this revised version authors made a significant effort to improve the manuscript by adding new experiments and text edits. The rationale for some experiments (that I pointed at in my previous comments) was included. Additionally, main text was edited providing clarity and an overall accurate result assessment (in results and discussion sections).

I fully agree that this work largely supports the role of LUX in mediating plant defenses (SA and JA mediated), and the role of JA in regulating the clock function. However, I find less convincing the evidence in support of LUX as a central node for the crosstalk between clock and defense responses. LUX regulates the differential response to morning vs evening (in LL) after Pma infiltration (statistical analysis in 2C should compare morning vs evening infection for each genotype) but not when Pma is sprayed (2B). Regulation of stomatal aperture by LUX (2D) provides a potential but not definitive explanation for these contrasting results. Morning and evening (in LL) Botritis infections (JA defenses) result in overall similar phenotypes in lux mutants (4D) suggesting that the clock does not regulate this defense response. On the other hand, the role of LUX in the regulation of clock responses to JA treatment was not really established, therefore it is unclear if JA regulates the clock by LUX regulation of JA signaling or by an independent mechanism.

These points should be properly addressed in the manuscript text.

Reviewer #4 (Remarks to the Author):

Revised manuscript provides the evidence that JA-Ile used in figure 5 did not cause growth retardation (Figure S8), but JA-Ile drastically decreased amplitude and slightly lengthened period. The effect of JA-Ile on clock parameters was completely canceled in the coi1-17 mutant. All data support the authors' idea, JA-Ile affects clock activity.

RESPONSE TO REVIEWERS' COMMENTS:

Reviewer #1 (Remarks to the Author):

The revised manuscript by Zhang et al. has addressed my previous concerns.

Reviewer #3 (Remarks to the Author):

In this revised version authors made a significant effort to improve the manuscript by adding new experiments and text edits. The rationale for some experiments (that I pointed at in my previous comments) was included. Additionally, main text was edited providing clarity and an overall accurate result assessment (in results and discussion sections).

I fully agree that this work largely supports the role of LUX in mediating plant defenses (SA and JA mediated), and the role of JA in regulating the clock function. However, I find less convincing the evidence in support of LUX as a central node for the crosstalk between clock and defense responses. LUX regulates the differential response to morning vs evening (in LL) after Pma infiltration (statistical analysis in 2C should compare morning vs evening infection for each genotype) but not when Pma is sprayed (2B). Regulation of stomatal aperture by LUX (2D) provides a potential but not definitive explanation for these contrasting results. Morning and evening (in LL) Botritis infections (JA defenses) result in overall similar phenotypes in lux mutants (4D) suggesting that the clock does not regulate this defense response. On the other hand, the role of LUX in the regulation of clock responses to JA treatment was not really established, therefore it is unclear if JA regulates the clock by LUX regulation of JA signaling or by an independent mechanism. These points should be properly addressed in the manuscript text.

According to suggestions from this reviewer and the editor, we have toned down the role of LUX in regulating the crosstalk between the circadian clock and plant defense. We have carefully checked through the manuscript and made corresponding changes. We have also included statistical analysis in figure 2C to show significant difference between morning and evening infection for Col-0.

Reviewer #4 (Remarks to the Author):

Revised manuscript provides the evidence that JA-Ile used in figure 5 did not cause growth retardation (Figure S8), but JA-Ile drastically decreased amplitude and slightly lengthened period. The effect of JA-Ile on clock parameters was completely canceled in the coi1-17 mutant. All data support the authors' idea, JA-Ile affects clock activity.